# TOPSIS Decision on Approximate Pareto Fronts by Using Evolutionary Algorithms: Application to an Engineering Design Problem

**Máximo Méndez** [1,*,†] **, Mariano Frutos** [2,†] **, Fabio Miguel** [3,†] **and Ricardo Aguasca-Colomo** [1,†]

1   Instituto Universitario SIANI, Universidad de Las Palmas de Gran Canaria (ULPGC),
    35017 Las Palmas de G.C., Spain; ricardo.aguasca@ulpgc.es
2   Department of Engineering, Universidad Nacional del Sur and CONICET, Bahía Blanca 8000, Argentina;
    mfrutos@uns.edu.ar
3   Universidad Nacional de Río Negro, Sede Alto Valle y Valle Medio, Villa Regina 8336, Argentina;
    fmiguel@unrn.edu.ar
*   Correspondence: maximo.mendez@ulpgc.es; Tel.: +34-928-458-702
†   These authors contributed equally to this work.

**Abstract:** A common technique used to solve multi-objective optimization problems consists of first generating the set of all Pareto-optimal solutions and then ranking and/or choosing the most interesting solution for a human decision maker (DM). Sometimes this technique is referred to as generate first–choose later. In this context, this paper proposes a two-stage methodology: a first stage using a multi-objective evolutionary algorithm (MOEA) to generate an approximate Pareto-optimal front of non-dominated solutions and a second stage, which uses the Technique for Order Preference by Similarity to an Ideal Solution (TOPSIS) devoted to rank the potential solutions to be proposed to the DM. The novelty of this paper lies in the fact that it is not necessary to know the ideal and nadir solutions of the problem in the TOPSIS method in order to determine the ranking of solutions. To show the utility of the proposed methodology, several original experiments and comparisons between different recognized MOEAs were carried out on a welded beam engineering design benchmark problem. The problem was solved with two and three objectives and it is characterized by a lack of knowledge about ideal and nadir values.

**Keywords:** multiple criteria decision-making; TOPSIS; preferences; engineering design; optimization; multi-objective evolutionary algorithms

## 1. Introduction

When real Multi-objective Optimization Problems (MOPs) are tackled, two different working approaches can be identified in the literature. The first, known as Multiple Criteria Decision-Making (MCDM) [1–13], is essentially interested in decision-making, for example in helping a human Decision-Maker (DM) to choose between various alternatives or solutions in accordance with several conflicting criteria or objectives. The main representatives of this approach can be found in schools of economics, management and finance and the role and participation of the DM before and during the decision-making process are decisive. The second, Multi-Objective Optimization (MOO) [14–19], more to the taste of engineers and mathematicians, is related to highly complex optimization problems, where, rather than the decision, the major interest lies in using fast algorithms to find a non-dominated set of solutions or Pareto-soptimal Front (POF). In this approach, DM participation in the search process may not be necessary. MCDM and MOO are therefore two disciplines belonging to two different

scientific communities, who solve similar problems and communicate with one another but have different competences.

The population-based Multi-Objective Evolutionary Algorithms (MOEA) [20–31], rather popular among the MOO scientific community, have shown a remarkable performance when solving hard optimization problems. These algorithms do not guarantee the determination of the exact POF, but the result is very close to the exact solution. Most MOEAs are categorized as a posteriori preference articulation, also referred to as Generate First–Choose Later (GFCL) [27,32]. The idea involves first generating multiple optimal Pareto solutions followed by choosing the most preferred one according to some criteria. The Non-Dominated Sorting Genetic Algorithm-II (NSGA-II) [33], the Multi-Objective Evolutionary Algorithm based on Decomposition (MOEA/D) [34] and the Global Weighting Achievement Scalarizing Function Genetic Algorithm (GWASF-GA) [35], to cite only three of the many relevant MOEAs, are recognized algorithms in the multi-objective literature that use this approach. NSGA-II is based on Pareto's dominance as a criterion to converge to the POF and crowding-distance operator as increasing diversification in the population. MOEA/D uses a strategy of decomposing the MOP into several scalar sub-problems that are simultaneously solved by the evolution of a population of solutions. GWASF-GA incorporates the ideas of NSGA-II and MOEA/D and it classifies solutions on Pareto fronts but based on the achievement scalarizing function of Wierzbicki [36]. On the other hand, Branke [37] suggested that, if a DM has some idea about what solutions to the problem might be preferred, this knowledge should be exploited. In this line, Branke proposed the integration of this imprecise knowledge (partial user preferences) in a MOEA, with the purpose of focusing the search for solutions in that region of the POF that is most relevant for the DM. The final result of this approach is a small region of the POF which contains the most likely preferred solutions for DM and from which the DM will select a solution. This approach also assumes a GFCL methodology and some examples that include DM's partial-preferences as a reference point are reported in [38–44]. The Non-g-Dominated Sorting Genetic Algorithm (g-NSGA-II in this work) modifies the dominance of Pareto in the original NSGA-II because of the g-dominance relation proposed in [41]. The Weighting Achievement Scalarizing Function Genetic Algorithm (WASF-GA) [43], similar to NSGA-II, devises the population of individuals into several fronts but based on the achievement scalarizing function of Wierzbicki [36] for each vector of weights in a sample of the weight vector space.

MOEAs have extensive applications in the engineering field [45] and, some of them, propose a two-stage methodology. The first stage, using some evolutionary method, is dedicated to building the best POF of solutions. The second stage engages some MCDM technique to select the most attractive one. This methodology has shown excellent potential in various optimization problems. In [46], a two-stage approach for solving multi-objective system reliability optimization problems is proposed. A POF is initially identified at the first stage by applying a MOEA. Quite often there are a large number of Pareto optimal solutions, and it is difficult, if not impossible, to effectively choose the representative solutions for the overall problem. To overcome this challenge, an integrated multi-objective selection optimization (MOSO) method is used in the second stage. In [47], a procedure to solve the multi-objective reactive power compensation problem is proposed. This procedure is based on the combination of a genetic algorithm (GA) and the $\epsilon$-dominance concept. Moreover, to help the DM to extract the best compromise solution from a finite set of alternatives the Technique for Order Preference by Similarity to an Ideal Solution (TOPSIS) is used. In [48], an approach integrating NSGA-II and TOPSIS method to optimize stochastic computer networks is proposed. NSGA-II searches for the POF where network reliability is evaluated in terms of minimal paths and recursive sum of disjoint products. Subsequently, TOPSIS method determines the best compromise solution. In [49], a hybrid method integrating Artificial Neural Network (ANN), modified NSGA-II and TOPSIS method for determining the optimum biodiesel blends and speed ranges of a diesel engine fueled with castor oil biodiesel blends is presented. First, an ANN predicts brake power, brake specific fuel consumption and the emissions of engine. Then, the modified NSGA-II is used for the multi-objective optimization process. Finally, an approach based on TOPSIS method is implemented for finding

the best compromise solution from the POF. In [50], a two-phase evaluation method is proposed for focusing on the characteristics of dynamic risk and multi attributes in project operations. In the first phase, a Markov process is used to evaluate the risk. Then, through the application of the TOPSIS method, a risk management strategy is selected considering completion time, cost, quality and probability of success as desired criteria. In [51], a hybrid approach integrating modified NSGA-II and TOPSIS method is proposed for achieving a lightweight design of the front sub-frame of a passenger car. Initially, the modified NSGA-II is employed for multi-objective optimization of the sub-frame, and then, by means of entropy weight theory and TOPSIS method, all the obtained solutions are ranked from the best to the worst in order to determine the best compromise solution. In [52], a decision-making tool based on multi-objective optimization technique MOORA is proposed. MOORA helps the designer for extracting the operating point as the best compromise solution to execute the candidate engineering design. In [53], an extended model predictive control scheme, called Multi-Objective Model Predictive Control (MOMPC), is described for dealing with real-time operation of a multi-reservoir system. The MOMPC approach incorporates a NSGA-II, MCDM and the receding horizon principle to solve a multi-objective reservoir operation problem in real time.

This paper proposes a methodology that follows a two-stage MOO+MCDM procedure. In the MOO stage (GF), a MOEA (any metaheuristic or deterministic method could have been used) obtains an approximate POF of solutions. Then, in the MCDM stage (CL), the $L_1$ distance metric is proposed and used in TOPSIS method (although another methodology supporting DM could have been used) in order to automatically obtain an approximate ranking of the solutions that could be interesting to a DM. The novelty of this work lies in the following aspects: (i) the decision is formulated based on an approximate POF of non-dominated solutions and consequently the ideal and nadir solutions to the real MOP may not be known; and (ii) even when the ideal and nadir solutions of the MOP are unknown, it is demonstrated in this work that, by using the $L_1$ distance metric in TOPSIS method, the best approximate ranking of solutions can be generated. In this context, no references (that we know of) indicate whether the ideal and nadir solutions, used in TOPSIS, are the true solutions of the MOP under study. The effectiveness of the proposed technique is verified by numerous experiments and performance comparisons between various MOEAs on a welded beam engineering design benchmark problem. Minimization of cost of fabrication, deflection and normal stress are the goals. This problem is characterized by a lack of knowledge about ideal and nadir values [54].

This article is structured as follows. The next section briefly explains some multi-objective basic concepts that make it easier to understand the work presented here. Section 3 details the proposed methodology. Section 4 gives application cases to validate the proposed method and lastly, Section 5 contains the conclusions.

## 2. Basic Concepts

Some basic definitions closely related to this study on MOO and MCDM are put forward in this section.

A MOP in terms of minimization is formalized as follows:

$$\begin{aligned} Min. \ & f(\mathbf{x}) = f_1(\mathbf{x}), \ldots, f_j(\mathbf{x}), \ldots, f_m(\mathbf{x}) \\ s.t. \ & \mathbf{x} \in X \end{aligned} \tag{1}$$

where $\mathbf{x} = (x_1, \ldots x_l, \ldots, x_k)$ is the decision variable vector, $X$ is the set of feasible solutions in the decision space, $j = (1, 2, \ldots, m$ objectives) and $l = (1, 2, \ldots, k$ decision variables). To represent the set of solutions $\mathbf{x} \in X$ in the objective space, we define:

$$Z = \{z = (z_1, \ldots, z_j, \ldots, z_m) \in R^m : z_1 = f_1(x), \ z_j = f_j(x), \ z_m = f_m(x), \forall x \in X\} \tag{2}$$

in Equation (2), $Z$ is the set of feasible solutions in the objective space and $z \in Z$ is a solution vector (image of $x \in X$) in the objective space.

Pareto dominance. A solution $z^u = (z_1^u, \ldots, z_j^u, \ldots, z_m^u)$ dominates a solution $z^v = (z_1^v, \ldots, z_j^v, \ldots, z_m^v)$ if and only if $\forall j \in (1, 2, \ldots, m)$ $z_j^u \leq z_j^v$ *and* $\exists j \in (1, 2, \ldots, m)$ such that $z_j^u < z_j^v$. If there are no solutions which dominates $z_j^u$, then $z_j^u$ is non-dominated.

Pareto-optimal front POF. The set of all non-dominated solutions $z \in Z$ in the objective space is known as the Pareto-Optimal Front.

Ideal solution $I^+$. Let us assume that only true POF of solutions are taken into account. The solution with the best possible values for each of the objective functions $I^+ = (I_1^+, \ldots, I_j^+, \ldots, I_m^+)$ is known as the ideal solution, i.e., $I_j^+ = \min. f_j(\mathbf{x})$ (see Figure 1).

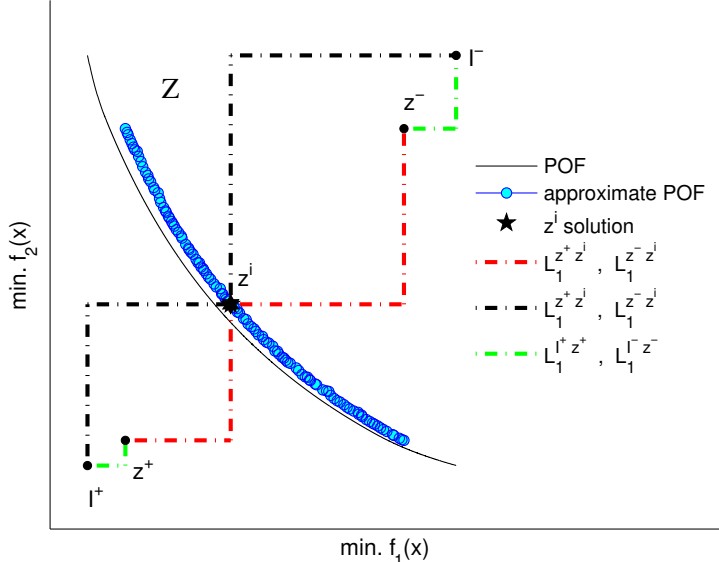

**Figure 1.** Feasible solution set $Z$ in the objectives space, the ideal $I^+$ and nadir $I^-$ solutions, the approximate ideal $z^+$ and nadir $z^-$ solutions and $L_1$ distances .

Nadir solution $I^-$. Let us assume that only true POF of solutions are taken into account. The solution with the worst possible values for each of the objective functions $z^- = (z_1^-, \ldots, z_j^-, \ldots, z_m^-)$ is known as the nadir solution, i.e., $z_j^- = \max. f_j(\mathbf{x})$ (see Figure 1).

Approximate ideal solution $z^+$. Let us assume that only approximate POF of solutions are taken into consideration. The solution with the best possible values for each of the objective functions $z^+ = (z_1^+, \ldots, z_j^+, \ldots, z_m^+)$ is known as the approximate-POF-based ideal solution, i.e., $z_j^+ = \min. f_j(\mathbf{x})$ (see Figure 1).

Approximate nadir solution $z^-$. Let us assume that only approximate POF of solutions are taken into account. The solution with the worst possible values for each of the objective functions $z^- = (z_1^-, \ldots, z_j^-, \ldots, z_m^-)$ is known as the approximate-POF-based nadir solution, i.e., $z_j^- = \min. f_j(\mathbf{x})$ (see Figure 1).

TOPSIS method. The TOPSIS method [2] establishes that the chosen solution should have the shortest distance to the ideal solution $I^+$ and the longest distance from the nadir solution $I^-$. The weighted distance of each solution from $I^+$ and $I^-$, according the chosen value $p$, can be, respectively, calculated as (3) and (4). Afterwards, the similarity ratio $S(z)$, defined in Equation (5), is assigned to each solution. The final ranking of solutions is obtained sorting the set of solutions decreasingly in terms of $S(z)$.

$$L_p^{I^+}(z) = \left[ \sum_{j=1}^{m} w_j^p \left| z_j - I_j^+ \right|^p \right]^{1/p} \tag{3}$$

$$L_p^{I^-}(z) = \left[ \sum_{j=1}^{m} w_j^p \left| I_j^- - z_j \right|^p \right]^{1/p} \tag{4}$$

$$S(z) = \frac{L_p^{I^-}(z^i)}{L_p^{I^+}(z^i) + L_p^{I^-}(z^i)} \quad 0 \le S(z) \le 1 \tag{5}$$

## 3. Methodology

The proposed method in this article draws together two independent technical stages of MOO and MCDM, as shown in Figure 2.

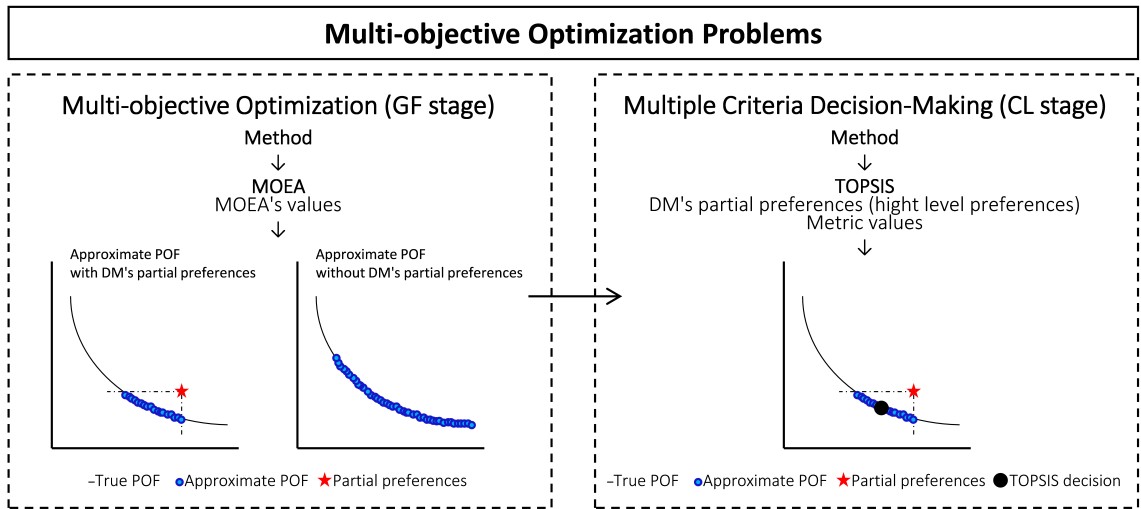

**Figure 2.** Proposed two-stage MOO and MCDM methodology.

Let us assume a MOP defined according to (1). Firstly, in the optimization stage (GF): (i) the DM decide which method to select to solve the MOP; and (ii) the DM specifies the parameters values of the metaheuristic algorithm used. Then, the algorithm is executed until the stopping condition has been reached. At this point, a discretized approximate POF of non-dominated solutions is available (see Figure 1) (two objective functions are considered), and the matrix formulation (6) where the POF=$\{z^i, i = 1, 2, \ldots, n\}$ of solutions is compared to the set of objective functions $\{z_j, j = 1, 2, \ldots, m\}$ according to the $e_{ij}$ evaluations of the solution $z^i$ regarding the objective $z_j$.

$$
\begin{array}{c}
\begin{array}{ccccccc}
z_1 & . & . & z_j & . & . & z_m
\end{array} \\
\begin{array}{c}
z^1 \\ . \\ . \\ z^i \\ . \\ . \\ z^n
\end{array}
\begin{pmatrix}
e_1^1 & . & . & e_j^1 & . & . & e_m^1 \\
. & . & . & . & . & . & . \\
. & . & . & . & . & . & . \\
e_1^i & . & . & e_j^i & . & . & e_m^i \\
. & . & . & . & . & . & . \\
. & . & . & . & . & . & . \\
e_1^n & . & . & e_j^n & . & . & e_m^n
\end{pmatrix}
\end{array} \tag{6}
$$

Subsequently, now in the decision-making stage (CL), we proceed as follows: (i) the DM decides which method to select for choosing the preferred solution; (ii) the DM expresses weights (high-level preferences) $w = \{w_j, j = 1, 2, \ldots, m\}$ associated to each objective function and metric value; (iii) based on the discretized approximate POF of obtained solutions, the weighted distance $L_1^{z^+}$ to the approximate-POF-based ideal solution $z^+$ and the weighted distance $L_1^{z^-}$ to the

approximate-POF-based nadir solution $z^-$ are computed for each solution (see Figure 1); and (iv) the final ranking of solutions is given by similarity ratio $S^*(z^i)$ defined in Equation (7).

$$S^*(z^i) = \frac{L_p^{z^-}(z^i)}{L_p^{z^+}(z^i) + L_p^{z^-}(z^i)} \tag{7}$$

**Proposition 1.** *Using the $L_1$ distance metric to the approximate-POF-based ideal ($z^+$) and nadir ($z^-$) solutions and the $L_1$ distance metric to the ideal ($I^+$) and nadir ($I^-$) solutions in TOPSIS method, $\forall\, z^i \in$ approximate POF, we obtain the same ranking of solutions.*

**Proof of Proposition 1.** Let us assume that the ideal $I^+ = (I_1^+, \ldots, I_m^+)$ and nadir $I^- = (I_1^-, \ldots, I_m^-)$ solutions are the true solutions of a real MOP. Using the $L_1$ distance in TOPSIS method, the ranking of solutions $z^i \in$ approximate POF can be calculated by solving (8). □

$$S(z^i) = \frac{L_1^{I^-}(z^i)}{L_1^{I^+}(z^i) + L_1^{I^-}(z^i)} \tag{8}$$

We consider now the distance between a solution $z^i$ and the $z^-, z^+, I^-, I^+$ solutions defined in Equation (9)–(12) and the distance between the $z^-, I^-$ solutions defined in Equation (13).

$$L_1^{z^-}(z^i) = \sum_{j=1}^{m} w_j \left| z_j^- - e_j^i \right| = \sum_{j=1}^{m} w_j (z_j^- - e_j^i) = \sum_{j=1}^{m} w_j z_j^- - \sum_{j=1}^{m} w_j e_j^i \tag{9}$$

$$L_1^{z^+}(z^i) = \sum_{j=1}^{m} w_j \left| e_j^i - z_j^+ \right| = \sum_{j=1}^{m} w_j (e_j^i - z_j^+) = \sum_{j=1}^{m} w_j e_j^i - \sum_{j=1}^{m} w_j z_j^+ \tag{10}$$

$$L_1^{I^-}(z^i) = \sum_{j=1}^{m} w_j \left| I_j^- - e_j^i \right| = \sum_{j=1}^{m} w_j (I_j^- - e_j^i) = \sum_{j=1}^{m} w_j I_j^- - \sum_{j=1}^{m} w_j e_j^i \tag{11}$$

$$L_1^{I^+}(z^i) = \sum_{j=1}^{m} w_j \left| e_j^i - I_j^+ \right| = \sum_{j=1}^{m} w_j (e_j^i - I_j^+) = \sum_{j=1}^{m} w_j e_j^i - \sum_{j=1}^{m} w_j I_j^+ \tag{12}$$

$$L_1^{\overline{z^- I^-}} = \sum_{j=1}^{m} w_j \left| I_j^- - z_j^- \right| = \sum_{j=1}^{m} w_j (I_j^- - z_j^-) = \sum_{j=1}^{m} w_j I_j^- - \sum_{j=1}^{m} w_j z_j^- = C_1 \tag{13}$$

where $C_1$ is a constant.

Furthermore, when $\sum_{j=1}^{m} w_j e_j^i$ is isolated and solved in (9) and subsequently replaced in (11), Equation (14) is obtained.

$$L_1^{I^-}(z^i) = L_1^{z^-}(z^i) - \sum_{j=1}^{m} w_j z_j^- + \sum_{j=1}^{m} w_j I_j^- \tag{14}$$

If we now consider Equations (13) and (14), we obtain (15).

$$L_1^{I^-}(z^i) = L_1^{z^-}(z^i) + C_1 \tag{15}$$

On the other hand, if we consider Equations (11) and (12), we have:

$$L_1^{I^+}(z^i) + L_1^{I^-}(z^i) = \sum_{j=1}^{m} w_j I_j^- - \sum_{j=1}^{m} w_j I_j^+ = C_2 \tag{16}$$

where $C_2$ is a constant.

Finally, if we take into account Equations (15) and (16) and subsequently replace them in (8), we obtain (17), which implies $\forall \, z^i \in$ approximate POF. Using the $L_1$ distance to the approximate-POF-based ideal ($z^+$) and nadir ($z^-$) solutions and the $L_1$ distance to the ideal ($I^+$) and nadir ($I^-$) solutions in TOPSIS method, we obtain the same ranking of solutions, even when said ideal $I^+$ and nadir $I^-$ solutions are unknown. $\square$

$$S(z^i) = \frac{L_1^{I^-}(z^i)}{L_1^{I^+}(z^i) + L_1^{I^-}(z^i)} = \frac{L_1^{z^-}(z^i) + C_1}{C_2} \tag{17}$$

To make using Equations (9)–(17) more intuitive, Figure 3 illustrates the distances $\overline{z^i z^-}$, $\overline{z^i I^-}$ and $\overline{z^- I^-}$ with $p = 1, 2, \infty$ metric. Assuming that nadir $I^-$ solution is known, it can be seen that Equation (15) is satisfied (and therefore proposition 1) if, and only if $p = 1$ metric is used in Equations (9)–(17), find below the details of the calculation. In addition, note that, using any of the other Pareto front $z^i$ solutions, the results are similar and distance $\overline{z^- I^-}$ is a constant ($C_1$).

$$p = 1 : L_1^{\overline{z^i z^-}} + L_1^{\overline{z^- I^-}} = (3 + 2) + (1 + 4) = 10 = L_1^{\overline{z^i I^-}} = (4 + 6) = 10$$
$$p = 2 : L_2^{\overline{z^i z^-}} + L_2^{\overline{z^- I^-}} = \sqrt{(3^2 + 2^2)} + \sqrt{(1^2 + 4^2)} = 7.73 \neq L_2^{\overline{z^i I^-}} = \sqrt{(4^2 + 6^2)} = 7.21$$
$$p = \infty : L_\infty^{\overline{z^i z^-}} + L_\infty^{\overline{z^- I^-}} = 3 + 4 = 7 \neq L_\infty^{\overline{z^i I^-}} = 6$$

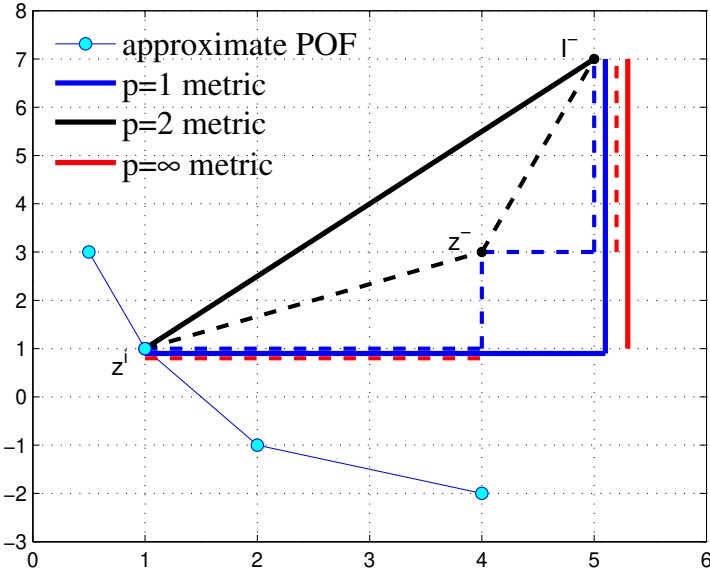

**Figure 3.** Distances $\overline{z^i z^-}$, $\overline{z^i I^-}$ and $\overline{z^- I^-}$ with $p = 1, 2, \infty$ metric.

Consequently, in convex problems, the method may be useful in providing an important clue to a DM in his/her final decision, especially when the true Pareto front of a multi-objective real-world problem is not available.

Finally, it should be highlighted that, in the proof of Proposition 1, Equations (9)–(16), all the sums go up to a value $m$ (objectives) and the methodology is therefore clearly applicable (at stage MCDM) in many objective optimization problems.

## 4. Results

In this section, we first apply the proposed methodology in this study to the bi-objective welded beam design problem. The objectives of the design are to minimize the cost of fabrication and to minimize the deflection. This problem is well-studied in both mono- [55–57] and multi-objective [52,54,58–60] literature. In the optimization stage, the NSGA-II, GWASF-GA and MOEA/D algorithms and the g-NSGA-II and WASF-GA algorithms (that include DM's partial-preferences as a reference point) were implemented with binary coding. In addition, tournament selection, uniform crossover and bitwise

mutation were used. The crossover probability was set to 0.8 and the mutation rate to $1/n$ where $n = 120$ is the string length; each variable (four design variables) uses 30 bits (eight decimal place precision). A population of sizes of $N = 50$ and $N = 100$ individuals and a maximum number of $G = 100$ generations were used. The hypervolume metric presented in [61] was used as a comparison measure between algorithms (see Figure 4). The reference point considered for the calculation of hypervolume was $(100.0, 0.1)$, which guarantees that it is dominated by all the solutions generated at the end of the evolution of the algorithms. Besides, the best cost objective value obtained by the algorithms were compared in terms of statistical results and number of function evaluations (i.e., $NFEs = N \times G$). Each algorithm was independently run 100 times for each test instance in the same initial conditions, from a randomly generated population. In the decision-making stage, when solving Equations (7) and (8), and to avoid any influence from the scale of measurement chosen for the various objectives, the objectives were normalized [7] using the procedure $\frac{|z^i - min\ z^i|}{|max\ z^i - min\ z^i|}$ with $z^i \in$ approximate POF.

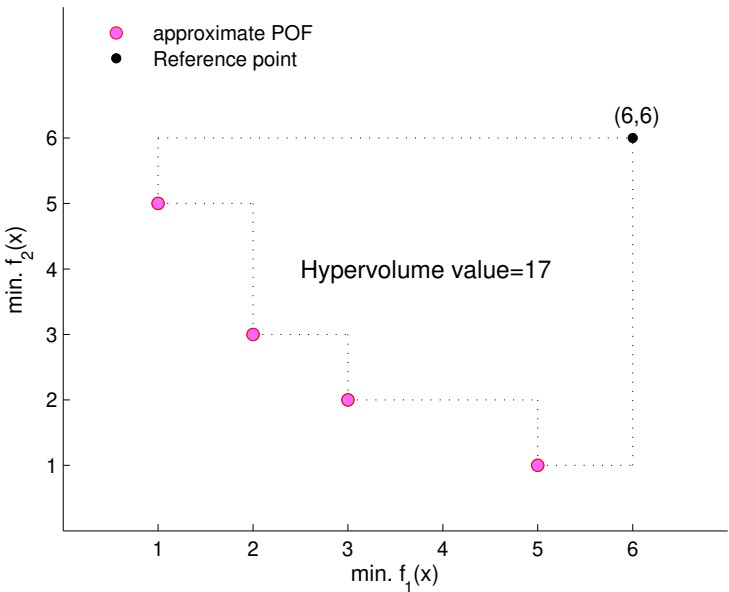

**Figure 4.** Evaluation of the hypervolume value with respect to the given reference point (6,6) on a two-objective minimization problem; larger hypervolume values indicate better quality of the approximate POF.

In a second test case, we demonstrate the utility of the suggested approach, by adding to the above mentioned problem the normal stress as a third objective function that should be minimized [54]. In this example, it was only considered the decision-making process. The TOPSIS [2] and ELECTRE I [6] methodologies were compared. In the TOPSIS and ELECTRE I approaches, equal weights values were assigned to all objective functions. Besides, a set of non-dominated solutions obtained in a randomized trial of NSGA-II ($N = 50$, $G = 500$) was used for comparisons.

### 4.1. Bi-Objective Welded Beam Design Problem (Optimization)

This design problem [58] minimizes both the cost and the deflection due to load $P$. The two objectives conflict since minimizing deflection will lead to an increase in manufacturing cost, which mainly includes the set-up cost, material cost and welding labor cost. The design involves four different design decision variables $(h, l, t, b)$ (see Figure 5) and four nonlinear constraints: shear stress,

normal stress, weld length and the buckling limitation. Formally, the bi-objective welded beam design problem can be defined as follows:

$$
\begin{aligned}
min. \quad & f_1(\mathbf{x}) = 1.10471h^2l + 0.04811tb(14.0 + l) \\
min. \quad & f_2(\mathbf{x}) = \delta(\mathbf{x}) = \frac{2.1952}{t^3b} \\
s.t. \quad & g_1(\mathbf{x}) = 13600 - \tau(\mathbf{x}) \geq 0 \\
& g_2(\mathbf{x}) = 30000 - \sigma(\mathbf{x}) \geq 0 \\
& g_3(\mathbf{x}) = b - h \geq 0 \\
& g_4(\mathbf{x}) = P_c(\mathbf{x}) - 6000 \geq 0 \\
& h, b \in [0.125, 5] \\
& l, t \in [0.1, 10] \\
where \quad &
\end{aligned}
\tag{18}
$$

$$
\tau(\mathbf{x}) = \sqrt{\frac{(\tau'(\mathbf{x}))^2 + (\tau''(\mathbf{x}))^2 + l\tau'(\mathbf{x})\tau''(\mathbf{x})}{\sqrt{0.25(l^2 + (h+t)^2)}}}
$$

$$
\tau'(\mathbf{x}) = \frac{6000}{\sqrt{2}hl}
$$

$$
\tau''(\mathbf{x}) = \frac{6000(14 + 0.5l)\sqrt{0.25(l^2 + (h+t)^2)}}{2\left[0.707hl(l^2/12 + 0.25(h+t)^2)\right]}
$$

$$
P_c(\mathbf{x}) = 64746.022(1 - 0.0282346t)tb^3
$$

$$
\sigma(\mathbf{x}) = \frac{504000}{t^2b}
$$

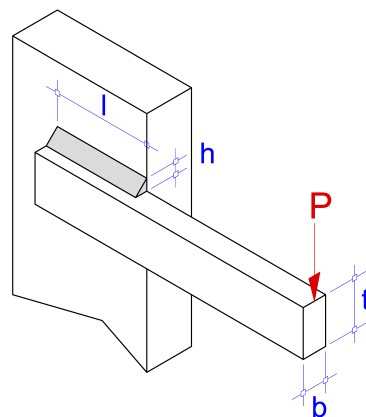

**Figure 5.** Welded beam design problem.

Firstly, NSGA-II, GWASF-GA and MOEA/D (other metaheuristic methods could have been used) were used to find the approximate Pareto fronts and then compared the results using the hypervolume metric. Table 1 shows the mean, standard deviation, best and worst hypervolume indicators achieved, over 100 independent runs. It can be seen that the NSGA-II and GWASF-GA algorithms attain the best performance. Figures 6 and 7 show a more detailed comparison between the algorithms from Table 1 with $N = 100$ and $G = 100$. Figure 6 shows the box plots based on the hypervolume approximation metric. We can see that the best median value and the lowest dispersion value are obtained with NSGA-II and GWASF-GA. In addition, in Figure 7 (left) that presents the evolution of the average hypervolume per generation and in Figure 7 (right) that shows the evolution of the standard deviation hypervolume, it is observed that the NSGA-II and GWASF-GA algorithms obtain similar values which are significantly better than the values achieved by MOEA/D.

**Table 1.** Comparison and statistical results of the mean, standard deviation values (upper), and the best and worst hypervolume values (lower), respectively, for NSGA-II, GWASF-GA and MOEA/D, over 100 runs.

|           | $N = 50$       | $N = 100$      |
| --------- | -------------- | -------------- |
| NSGA-II   | 9.4734–0.1142  | 9.5643–0.0830  |
|           | 9.6612–9.2145  | 9.6658–9.2703  |
| GWASF-GA  | 9.5067–0.1166  | 9.5717–0.0854  |
|           | 9.6653–9.1825  | 9.6721–9.2619  |
| MOEA/D    | 9.1202–0.3977  | 9.1455–0.3495  |
|           | 9.6594–7.7837  | 9.6549–8.3405  |

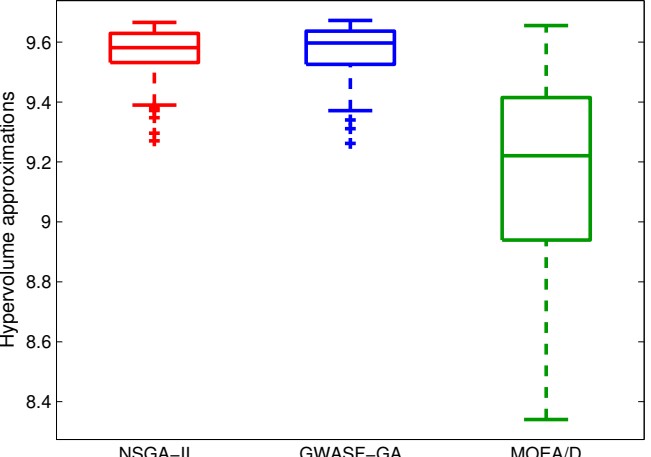

**Figure 6.** Box-plots based on the hypervolume metric for NSGA-II, GWASF-GA and MOEA/D ($N = 100$).

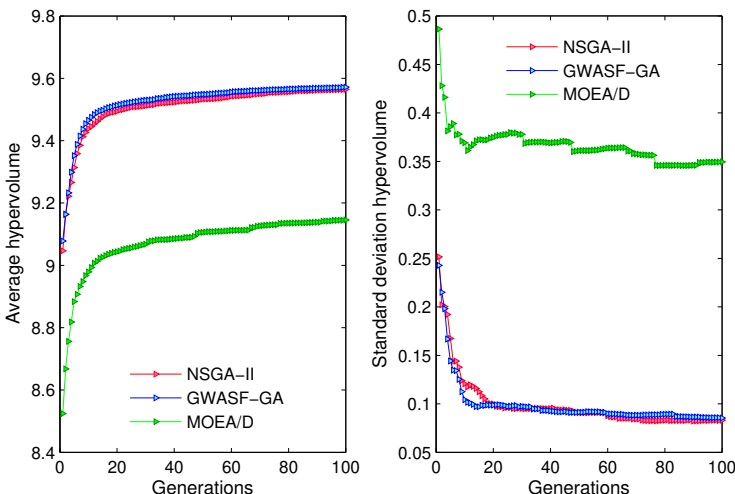

**Figure 7.** Evolution of the average hypervolume (**left**); and evolution of the standard deviation hypervolume (**right**) for NSGA-II, GWASF-GA and MOEA/D ($N = 100$).

Moreover, various MOEAs have been applied to the problem (18) by different researchers. Table 2 shows the statistical results of the best objective cost, the mean value and the standard deviation value obtained in this work with NSGA-II ($NFEs =$5000 and 10,000), GWASF-GA ($NFEs =$5000 and 10,000) and MOEA/D ($NFEs =$5000 and 10,000) with those attained by other bi-objective metaheuristics. It can see that GWASF-GA ($NFEs =$10,000) has the best objective cost followed

by NSGA-II ($NFEs$ =10,000) and MOWCA ($NFEs$ =15,000). In addition, it can be noted in Table 2 that GWASF-GA with $NFEs$ =10,000 ($N = 100$ and $G = 100$) reaches the best mean value (3.5657).

**Table 2.** Comparison and statistical results of the best objective cost, mean and standard deviation, respectively, found by different MOEAs (NA, not available).

| Algorithms | $NFEs$ | Best | Mean | Std. Dev. |
|---|---|---|---|---|
| NSGA-II [58] | 10,000 | 2.7900 | NA | NA |
| $pa\epsilon$-ODEMO [59] | 15000 | 2.8959 | NA | NA |
| MOWCA [60] | 15,000 | 2.5325 | NA | NA |
| M20-CSA [52] | 12,000 | 7.9669 | NA | NA |
| MOCCSA [52] | 12,000 | 13.6193 | NA | NA |
| MOCSA [52] | 12,000 | 3.6842 | NA | NA |
| NSGA-II Present study | 5000 | 2.5279 | 4.5480 | 1.2005 |
| NSGA-II Present study | 10,000 | 2.5257 | 3.6236 | 0.8807 |
| GWASF-GA Present study | 5000 | 2.5313 | 4.231306 | 1.2324 |
| GWASF-GA Present study | 10,000 | 2.4553 | 3.5657 | 0.9138 |
| MOEA/D Present study | 5000 | 2.5835 | 8.0708 | 4.0780 |
| MOEA/D Present study | 10,000 | 2.6263 | 7.8712 | 3.5982 |

Secondly, a similar study to the previous one using the g-NSGA-II and WASF-GA algorithms (although other metaheuristic methods with DM's partial-preferences could have been used) was performed considering three different reference points (DM's partial-preferences) $(4, 0.003)$, $(15, 0.0025)$ and $(30, 0.001)$, infeasible and feasible (see Figure 8). The hypervolume metric of the region of interest defined in [43] was used as a comparison measure for the two algorithms.

Table 3 presents the mean, standard deviation, best and worst hypervolume indicators achieved, over 100 independent runs, by the g-NSGA-II and WASF-GA algorithms. It can be perceived that the values obtained can be quite different depending on the reference point used. For example, when the reference point was set to $(4, 0.003)$ (non-feasible), the performances obtained for both, g-NSGA-II and WASF-GA, were similar (see also Figures 8–10). On the other hand, when the reference point was $(30, 0.001)$ (feasible), g-NSGA-II and WASF-GA also had similar results, although g-NSGA-II had a slightly better performance of the hypervolume metric (Figure 11 and Table 3) and better distribution of the approximate Pareto front's solution set (Figure 8). However, the values in Table 3 show that the WASF-GA algorithm obtained superior performance than g-NSGA-II when the reference point was set to $(15, 0.0025)$ (see also Figures 8, 9 and 12).

**Table 3.** Comparison and statistical results of the mean, standard deviation values (upper), and the best and worst hypervolume values (lower), respectively, for three different DM's partial-preferences $(4, 0.003)$, $(15, 0.0025)$ and $(30, 0.001)$ for g-NSGA-II and WASF-GA.

| | $N = 50$ | $N = 100$ | $N = 50$ | $N = 100$ | $N = 50$ | $N = 100$ |
|---|---|---|---|---|---|---|
| | $(4, 0.003)$ | | $(15, 0.0025)$ | | $(30, 0.001)$ | |
| g- NSGA-II | 4.420–0.079 | 4.451–0.032 | 3.833–0.297 | 3.988–0.264 | 3.059–0.293 | 3.177–0.103 |
| | 4.459–3.936 | 4.460–4.162 | 4.221–3.410 | 4.224–3.411 | 3.306–1.978 | 3.306–2.980 |
| WASF-GA | 4.382–0.135 | 4.450–0.020 | 4.085–0.116 | 4.156–0.075 | 3.065–0.050 | 3.101–0.031 |
| | 4.459–3.843 | 4.459–4.342 | 4.204–3.645 | 4.206–3.933 | 3.136–2.898 | 3.139–3.002 |

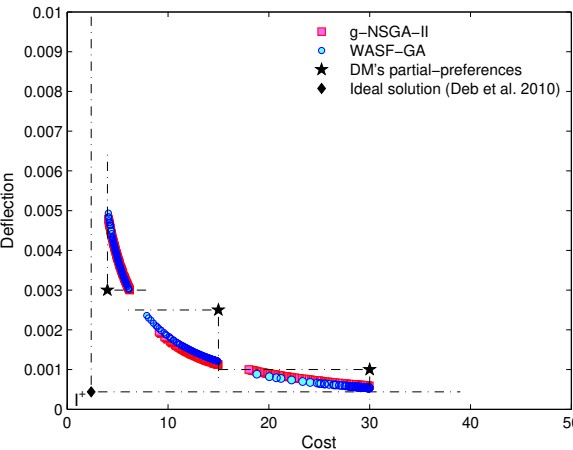

**Figure 8.** DM's partial-preferences $(4, 0.003)$, $(15, 0.0025)$ and $(30, 0.001)$ and the respective approximate POF with the hypervolume indicator closest to the average value of hypervolume after 100 runs for g-NSGA-II and WASF-GA ($N = 100$).

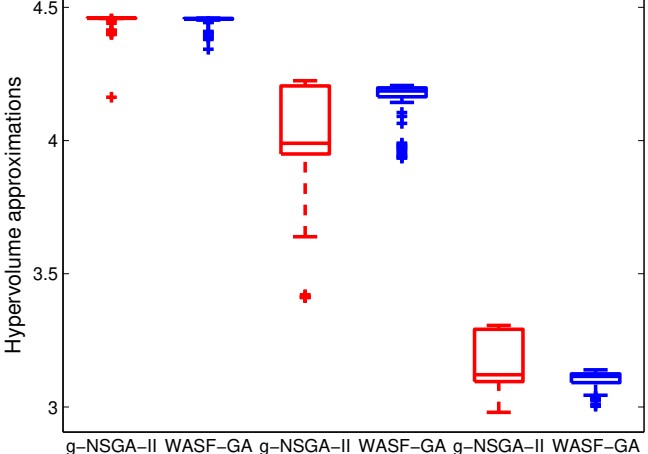

**Figure 9.** Box-plots based on the hypervolume metrics $(4, 0.003)$ (**left**), $(15, 0.0025)$ (**middle**) and $(30, 0.001)$ (**right**) for g-NSGA-II and WASF-GA ($N = 100$).

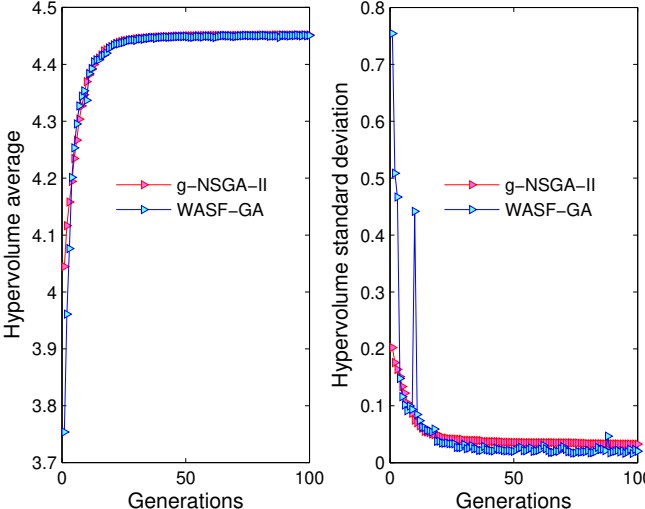

**Figure 10.** Evolution of the average hypervolume (**left**); and evolution of the standard deviation hypervolume (**right**) for g-NSGA-II, WASF-GA, DM's partial-preferences $(4, 0.003)$ ($N = 100$).

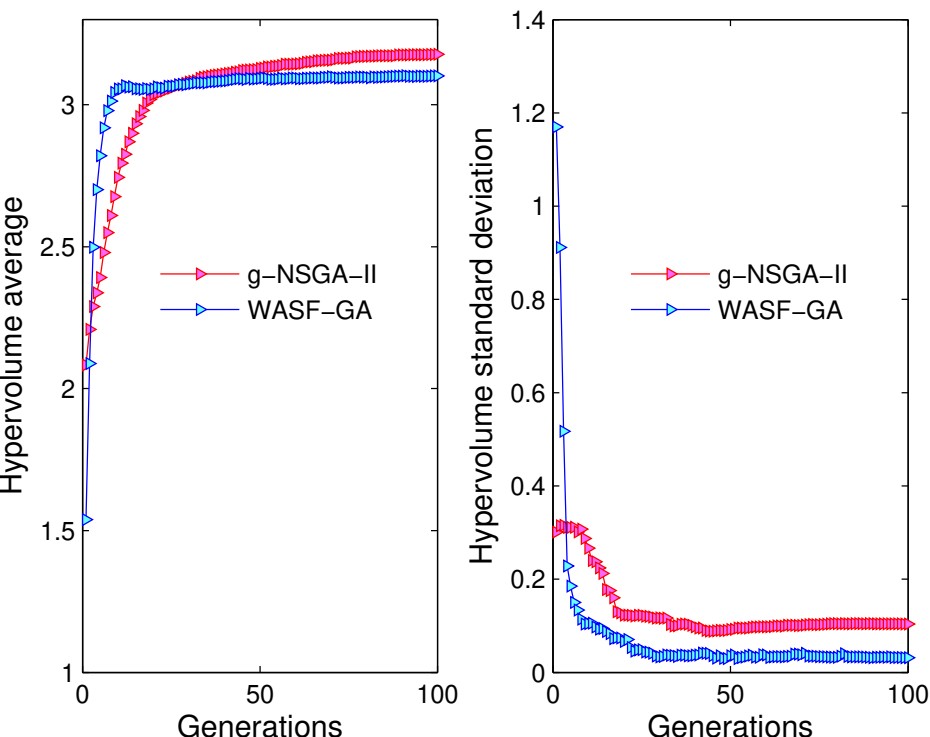

**Figure 11.** Evolution of the average hypervolume (**left**); and evolution of the standard deviation hypervolume (**right**) for g-NSGA-II, WASF-GA, DM's partial-preferences $(30, 0.001)$ $(N = 100)$.

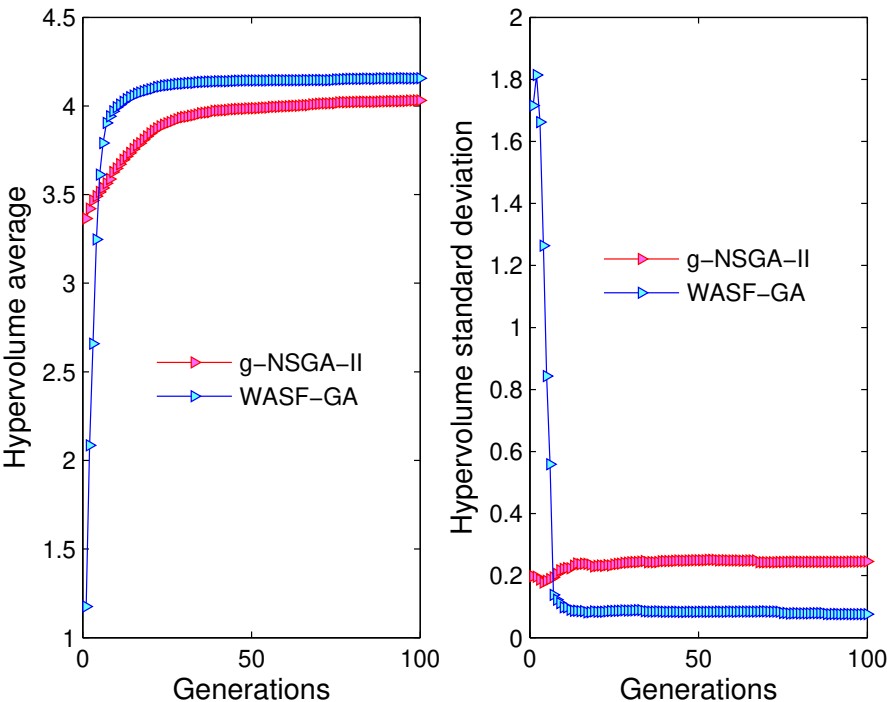

**Figure 12.** Evolution of the average hypervolume (**left**); and evolution of the standard deviation hypervolume (**right**) for g-NSGA-II, WASF-GA, DM's partial-preferences $(15, 0.0025)$ $(N = 100)$.

4.1.1. Bi-Objective Welded Beam Design Problem (Decision)

In this section, the second stage (CL) is executed. Now, TOPSIS (other method supporting DM could have been used) is used to rank the solutions and to determine the best TOPSIS decision (rank-1

solution). The $L_1$ and $L_2$ metrics in TOPSIS model were utilized. In addition, the approximate Pareto front with the hypervolume indicator closest to the average value of hypervolume after 100 runs was adopted for comparisons and for each algorithm (see Figure 13, left and right (with logarithmic scale to appreciate de nadir solution)). As expected, the appearance of the approximate set of Pareto optimal solutions changes with the trial and the employed algorithm. Therefore, the choice of the DM is conditioned by the quality of the POF achieved. The best known ideal and nadir values $(2.3810, 0.000439)$ and $(333.9095, 0.0713)$, respectively [54], of the problem (18) were used in the experiments (see Figure 13, right).

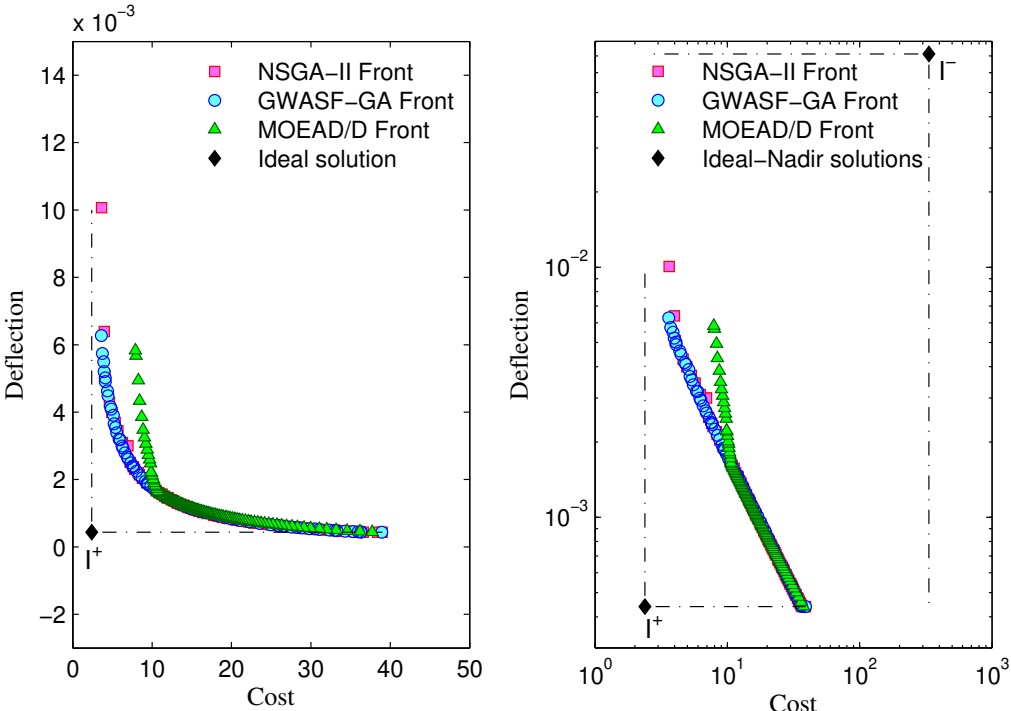

**Figure 13.** Approximate POF with the hypervolume indicator closest to the average value of hypervolume after 100 runs (**left**); and the same data drawn with logarithmic scale (**right**) for NSGA-II, GWASF-GA and MOEA/D ($N = 100$).

First, POF resulting from the NSGA-II, GWASF-GA and MOEA/D algorithms were considered. Tables 4 and 5 give the eight best solutions ranked from best to worst. The first two columns represent the coordinates of the solutions in the objective space, the third and fourth columns give the distances of the solutions regarding the $z^+$ and $z^-$ and the $I^+$ and $I^-$ solutions, the fifth column gives the similarity values $S^{z^+ z^-}$ and $S^{I^+ I^-}$ according TOPSIS method and the last two columns show the ranking of solutions regarding both the $z^+$ and $z^-$ and the $I^+$ and $I^-$ solutions. The results in Table 4 show that, by using $L_1$ metric in the TOPSIS model, with respect to both the ideal $z^+$ and nadir $z^-$ solutions of the approximate POF obtained by the algorithms and the $I^+$ and $I^-$ solutions of the real POM, the ranking of the proposal solutions bears the same ranking. However, when $L_2$ metric is used, the ranking of the proposed solutions differs (see the last two columns of the Table 5).

**Table 4.** TOPSIS ranking results with $L_1$ metric for NSGA-II, GWASF-GA and MOEA/D ($N = 100$).

| NSGA-II | Cost | Deflection | $L_1^{z^i z^+}$ | $L_1^{z^i z^-}$ | $S^{z^+ z^-}$ | Rank$^{z^+ z^-}$ | Rank$^{I^+ I^-}$ |
|---|---|---|---|---|---|---|---|
| | 9.3441 | 0.0019 | 0.0190 | 0.1023 | 0.8436 | 1 | 1 |
| | 8.7703 | 0.0020 | 0.0190 | 0.1022 | 0.8431 | 2 | 2 |
| | 10.520 | 0.0017 | 0.0191 | 0.1022 | 0.8428 | 3 | 3 |
| | 8.3866 | 0.0021 | 0.0192 | 0.1020 | 0.8418 | 4 | 4 |
| | 10.854 | 0.0016 | 0.0192 | 0.1020 | 0.8418 | 5 | 5 |
| | 9.7900 | 0.0018 | 0.0192 | 0.1020 | 0.8416 | 6 | 6 |
| | 8.9661 | 0.0020 | 0.0193 | 0.1019 | 0.8406 | 7 | 7 |
| | 8.2575 | 0.0022 | 0.0194 | 0.1018 | 0.8398 | 8 | 8 |

| NSGA-II | Cost | Deflection | $L_1^{z^i I^+}$ | $L_1^{z^i I^-}$ | $S^{I^+ I^-}$ | Rank$^{I^+ I^-}$ | Rank$^{z^+ z^-}$ |
|---|---|---|---|---|---|---|---|
| | 9.3441 | 0.0019 | 0.0208 | 0.9792 | 0.9792 | 1 | 1 |
| | 8.7703 | 0.0020 | 0.0209 | 0.9791 | 0.9791 | 2 | 2 |
| | 10.520 | 0.0017 | 0.0209 | 0.9791 | 0.9791 | 3 | 3 |
| | 8.3866 | 0.0021 | 0.0211 | 0.9789 | 0.9789 | 4 | 4 |
| | 10.854 | 0.0016 | 0.0211 | 0.9789 | 0.9789 | 5 | 5 |
| | 9.7900 | 0.0018 | 0.0211 | 0.9789 | 0.9789 | 6 | 6 |
| | 8.9661 | 0.0020 | 0.0212 | 0.9788 | 0.9788 | 7 | 7 |
| | 8.2575 | 0.0022 | 0.0213 | 0.9787 | 0.9787 | 8 | 8 |

| GWASF-GA | Cost | Deflection | $L_1^{z^i z^+}$ | $L_1^{z^i z^-}$ | $S^{z^+ z^-}$ | Rank$^{z^+ z^-}$ | Rank$^{I^+ I^-}$ |
|---|---|---|---|---|---|---|---|
| | 9.4910 | 0.0019 | 0.0189 | 0.0755 | 0.8002 | 1 | 1 |
| | 9.3520 | 0.0019 | 0.0189 | 0.0755 | 0.8002 | 2 | 2 |
| | 9.3810 | 0.0019 | 0.0189 | 0.0755 | 0.7999 | 3 | 3 |
| | 9.8331 | 0.0018 | 0.0189 | 0.0755 | 0.7993 | 4 | 4 |
| | 9.1265 | 0.0019 | 0.0189 | 0.0755 | 0.7993 | 5 | 5 |
| | 10.220 | 0.0017 | 0.0190 | 0.0754 | 0.7991 | 6 | 6 |
| | 10.521 | 0.0017 | 0.0191 | 0.0753 | 0.7980 | 7 | 7 |
| | 10.515 | 0.0017 | 0.0191 | 0.0753 | 0.7978 | 8 | 8 |

| GWASF-GA | Cost | Deflection | $L_1^{z^i I^+}$ | $L_1^{z^i I^-}$ | $S^{I^+ I^-}$ | Rank$^{I^+ I^-}$ | Rank$^{z^+ z^-}$ |
|---|---|---|---|---|---|---|---|
| | 9.4910 | 0.0019 | 0.0207 | 0.9793 | 0.9793 | 1 | 1 |
| | 9.3520 | 0.0019 | 0.0207 | 0.9793 | 0.9793 | 2 | 2 |
| | 9.3810 | 0.0019 | 0.0207 | 0.9793 | 0.9793 | 3 | 3 |
| | 9.8331 | 0.0018 | 0.0208 | 0.9792 | 0.9792 | 4 | 4 |
| | 9.1265 | 0.0019 | 0.0208 | 0.9792 | 0.9792 | 5 | 5 |
| | 10.220 | 0.0017 | 0.0208 | 0.9792 | 0.9792 | 6 | 6 |
| | 10.521 | 0.0017 | 0.0209 | 0.9791 | 0.9791 | 7 | 7 |
| | 10.515 | 0.0017 | 0.0209 | 0.9791 | 0.9791 | 8 | 8 |

| MOEA/D | Cost | Deflection | $L_1^{z^i z^+}$ | $L_1^{z^i z^-}$ | $S^{z^+ z^-}$ | Rank$^{z^+ z^-}$ | Rank$^{I^+ I^-}$ |
|---|---|---|---|---|---|---|---|
| | 10.754 | 0.0016 | 0.0127 | 0.0703 | 0.8471 | 1 | 1 |
| | 10.913 | 0.0016 | 0.0128 | 0.0703 | 0.8464 | 2 | 2 |
| | 10.689 | 0.0017 | 0.0128 | 0.0702 | 0.8462 | 3 | 3 |
| | 11.059 | 0.0016 | 0.0128 | 0.0702 | 0.8457 | 4 | 4 |
| | 11.236 | 0.0016 | 0.0129 | 0.0701 | 0.8446 | 5 | 5 |
| | 11.384 | 0.0015 | 0.0130 | 0.0701 | 0.8439 | 6 | 6 |
| | 11.571 | 0.0015 | 0.0131 | 0.0699 | 0.8423 | 7 | 7 |
| | 10.575 | 0.0017 | 0.0131 | 0.0699 | 0.8422 | 8 | 8 |

| MOEA/D | Cost | Deflection | $L_1^{z^i I^+}$ | $L_1^{z^i I^-}$ | $S^{z^+ z^-}$ | Rank$^{I^+ I^-}$ | Rank$^{z^+ z^-}$ |
|---|---|---|---|---|---|---|---|
| | 10.754 | 0.0016 | 0.0210 | 0.9790 | 0.9790 | 1 | 1 |
| | 10.913 | 0.0016 | 0.0211 | 0.9789 | 0.9789 | 2 | 2 |
| | 10.689 | 0.0017 | 0.0211 | 0.9789 | 0.9789 | 3 | 3 |
| | 11.059 | 0.0016 | 0.0211 | 0.9789 | 0.9789 | 4 | 4 |
| | 11.236 | 0.0016 | 0.0212 | 0.9788 | 0.9787 | 5 | 5 |
| | 11.384 | 0.0015 | 0.0213 | 0.9787 | 0.9787 | 6 | 6 |
| | 11.571 | 0.0015 | 0.0214 | 0.9786 | 0.9786 | 7 | 7 |
| | 10.575 | 0.0017 | 0.0214 | 0.9786 | 0.9786 | 8 | 8 |

**Table 5.** TOPSIS ranking results with $L_2$ metric for NSGA-II, GWASF-GA and MOEA/D ($N = 100$).

| NSGA-II | Cost | Deflection | $L_2^{z^i z^+}$ | $L_2^{z^i z^-}$ | $S^{z^+ z^-}$ | Rank$^{z^+ z^-}$ | Rank$^{I^+ I^-}$ |
|---|---|---|---|---|---|---|---|
| | 9.3441 | 0.0019 | 0.0190 | 0.1031 | 0.8441 | 1 | 1 |
| | 10.520 | 0.0017 | 0.0191 | 0.1035 | 0.8439 | 2 | 4 |
| | 9.7900 | 0.0018 | 0.0192 | 0.1030 | 0.8428 | 3 | 3 |
| | 10.854 | 0.0016 | 0.0194 | 0.1035 | 0.8424 | 4 | 7 |
| | 8.7703 | 0.0020 | 0.0193 | 0.1028 | 0.8416 | 5 | 2 |
| | 8.9661 | 0.0020 | 0.0196 | 0.1025 | 0.8396 | 6 | 5 |
| | 11.246 | 0.0016 | 0.0198 | 0.1034 | 0.8394 | 7 | 10 |
| | 8.3866 | 0.0021 | 0.0198 | 0.1025 | 0.8383 | 8 | 6 |

| NSGA-II | Cost | Deflection | $L_2^{z^i I^+}$ | $L_2^{z^i I^-}$ | $S^{I^+ I^-}$ | Rank$^{I^+ I^-}$ | Rank$^{z^+ z^-}$ |
|---|---|---|---|---|---|---|---|
| | 9.3441 | 0.0019 | 0.0208 | 0.9792 | 0.9792 | 1 | 1 |
| | 8.7703 | 0.0020 | 0.0210 | 0.9791 | 0.9790 | 2 | 5 |
| | 9.7900 | 0.0018 | 0.0211 | 0.9789 | 0.9789 | 3 | 3 |
| | 10.520 | 0.0017 | 0.0213 | 0.9791 | 0.9788 | 4 | 2 |
| | 8.9661 | 0.0020 | 0.0212 | 0.9788 | 0.9788 | 5 | 6 |
| | 8.3866 | 0.0021 | 0.0213 | 0.9789 | 0.9787 | 6 | 8 |
| | 10.854 | 0.0016 | 0.0215 | 0.9789 | 0.9785 | 7 | 4 |
| | 8.2575 | 0.0022 | 0.0216 | 0.9787 | 0.9784 | 8 | 11 |

| GWASF-GA | Cost | Deflection | $L_2^{z^i z^+}$ | $L_2^{z^i z^-}$ | $S^{z^+ z^-}$ | Rank$^{z^+ z^-}$ | Rank$^{I^+ I^-}$ |
|---|---|---|---|---|---|---|---|
| | 9.4910 | 0.0019 | 0.0189 | 0.0767 | 0.8023 | 1 | 2 |
| | 9.3520 | 0.0019 | 0.0189 | 0.0768 | 0.8022 | 2 | 1 |
| | 9.3810 | 0.0019 | 0.0189 | 0.0767 | 0.8020 | 3 | 3 |
| | 9.8331 | 0.0018 | 0.0189 | 0.0765 | 0.8014 | 4 | 5 |
| | 9.1265 | 0.0019 | 0.0191 | 0.0768 | 0.8010 | 5 | 4 |
| | 10.220 | 0.0017 | 0.0190 | 0.0763 | 0.8006 | 6 | 6 |
| | 10.521 | 0.0017 | 0.0192 | 0.0760 | 0.7988 | 7 | 9 |
| | 10.515 | 0.0017 | 0.0192 | 0.0760 | 0.7987 | 8 | 10 |

| GWASF-GA | Cost | Deflection | $L_2^{z^i I^+}$ | $L_2^{z^i I^-}$ | $S^{I^+ I^-}$ | Rank$^{I^+ I^-}$ | Rank$^{z^+ z^-}$ |
|---|---|---|---|---|---|---|---|
| | 9.3520 | 0.0019 | 0.0207 | 0.9793 | 0.9793 | 1 | 2 |
| | 9.4910 | 0.0019 | 0.0207 | 0.9793 | 0.9793 | 2 | 1 |
| | 9.3810 | 0.0019 | 0.0207 | 0.9793 | 0.9793 | 3 | 3 |
| | 9.1265 | 0.0019 | 0.0208 | 0.9792 | 0.9792 | 4 | 5 |
| | 9.8331 | 0.0018 | 0.0209 | 0.9792 | 0.9791 | 5 | 4 |
| | 10.220 | 0.0017 | 0.0210 | 0.9792 | 0.9790 | 6 | 6 |
| | 8.7400 | 0.0020 | 0.0210 | 0.9791 | 0.9790 | 7 | 9 |
| | 8.4192 | 0.0021 | 0.0212 | 0.9790 | 0.9788 | 8 | 11 |

| MOEA/D | Cost | Deflection | $L_2^{z^i z^+}$ | $L_2^{z^i z^-}$ | $S^{z^+ z^-}$ | Rank$^{z^+ z^-}$ | Rank$^{I^+ I^-}$ |
|---|---|---|---|---|---|---|---|
| | 11.059 | 0.0016 | 0.0132 | 0.0709 | 0.8429 | 1 | 5 |
| | 11.236 | 0.0016 | 0.0132 | 0.0708 | 0.8428 | 2 | 7 |
| | 11.384 | 0.0015 | 0.0132 | 0.0707 | 0.8427 | 3 | 9 |
| | 10.913 | 0.0016 | 0.0133 | 0.0711 | 0.8426 | 4 | 3 |
| | 10.754 | 0.0016 | 0.0133 | 0.0712 | 0.8422 | 5 | 1 |
| | 11.571 | 0.0015 | 0.0132 | 0.0705 | 0.8418 | 6 | 11 |
| | 11.724 | 0.0015 | 0.0133 | 0.0704 | 0.8415 | 7 | 12 |
| | 11.874 | 0.0015 | 0.0133 | 0.0703 | 0.8410 | 8 | 13 |

| MOEA/D | Cost | Deflection | $L_2^{z^i I^+}$ | $L_2^{z^i I^-}$ | $S^{I^+ I^-}$ | Rank$^{I^+ I^-}$ | Rank$^{z^+ z^-}$ |
|---|---|---|---|---|---|---|---|
| | 10.754 | 0.0016 | 0.0214 | 0.9790 | 0.9786 | 1 | 5 |
| | 10.689 | 0.0017 | 0.0215 | 0.9789 | 0.9785 | 2 | 9 |
| | 10.913 | 0.0016 | 0.0216 | 0.9789 | 0.9784 | 3 | 4 |
| | 10.575 | 0.0017 | 0.0217 | 0.9786 | 0.9783 | 4 | 14 |
| | 11.059 | 0.0016 | 0.0217 | 0.9789 | 0.9783 | 5 | 1 |
| | 10.469 | 0.0018 | 0.0219 | 0.9783 | 0.9781 | 6 | 18 |
| | 11.236 | 0.0016 | 0.0219 | 0.9788 | 0.9781 | 7 | 2 |
| | 10.387 | 0.0018 | 0.0220 | 0.9782 | 0.9780 | 8 | 21 |

Note that the best solution (rank-1) is referred to in this paper as the TOPSIS decision. Logically, this solution does not change when $L_1$ metric is used (see Figures 14–16 (left) and Table 4). Nevertheless, this is not always the case when using the metric $L_2$ (see Figures 14–16 (right) and Table 5).

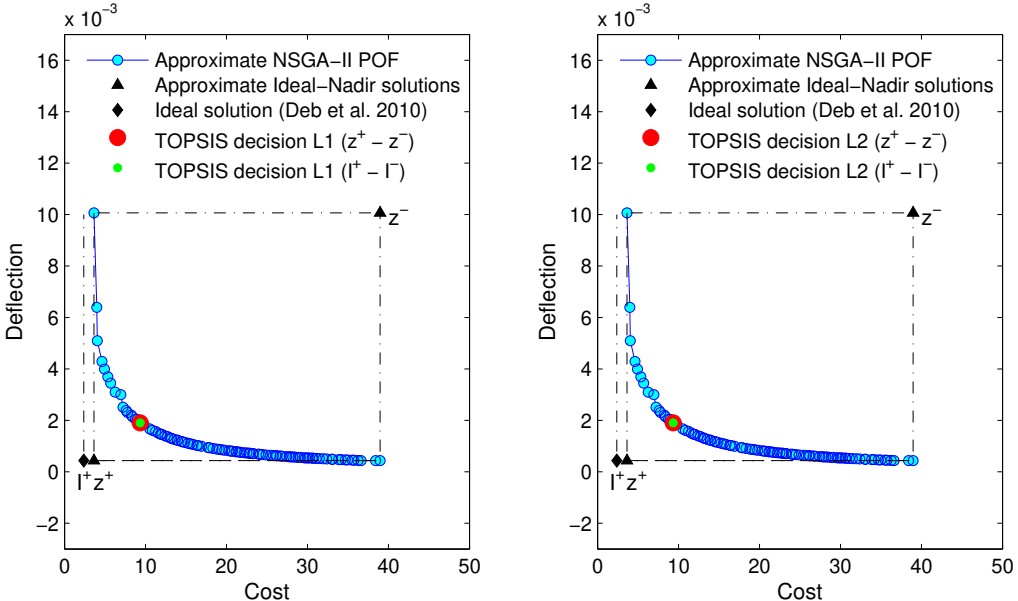

**Figure 14.** TOPSIS decision with $L_1$ (**left**) and $L_2$ (**right**) metrics on the approximate POF with the hypervolume indicator closest to the average value of hypervolume after 100 runs for NSGA-II ($N = 100$).

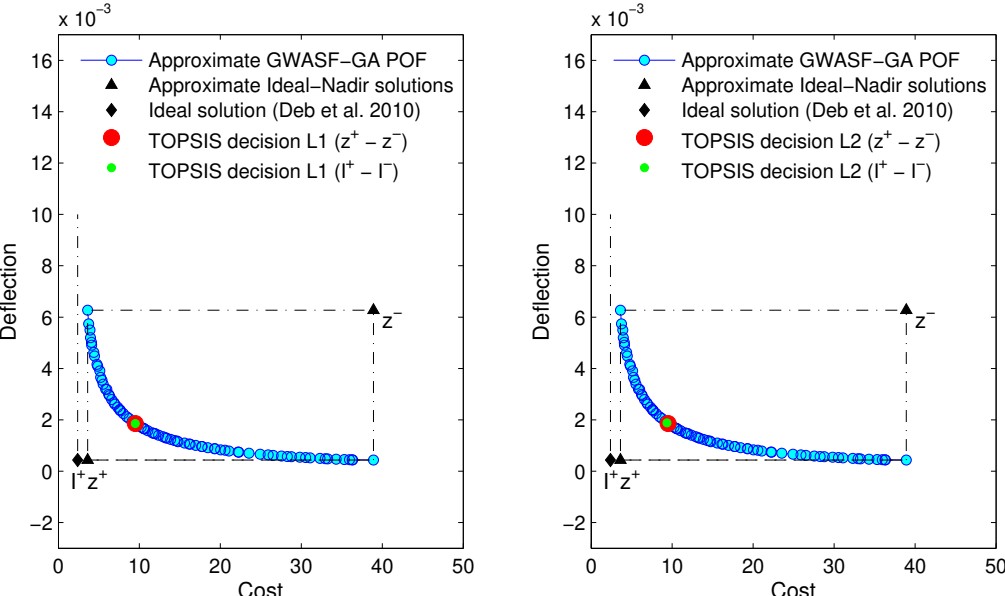

**Figure 15.** TOPSIS decision with $L_1$ (**left**) and $L_2$ (**right**) metrics on the approximate POF with the hypervolume indicator closest to the average value of hypervolume after 100 runs for GWASF-GA ($N = 100$).

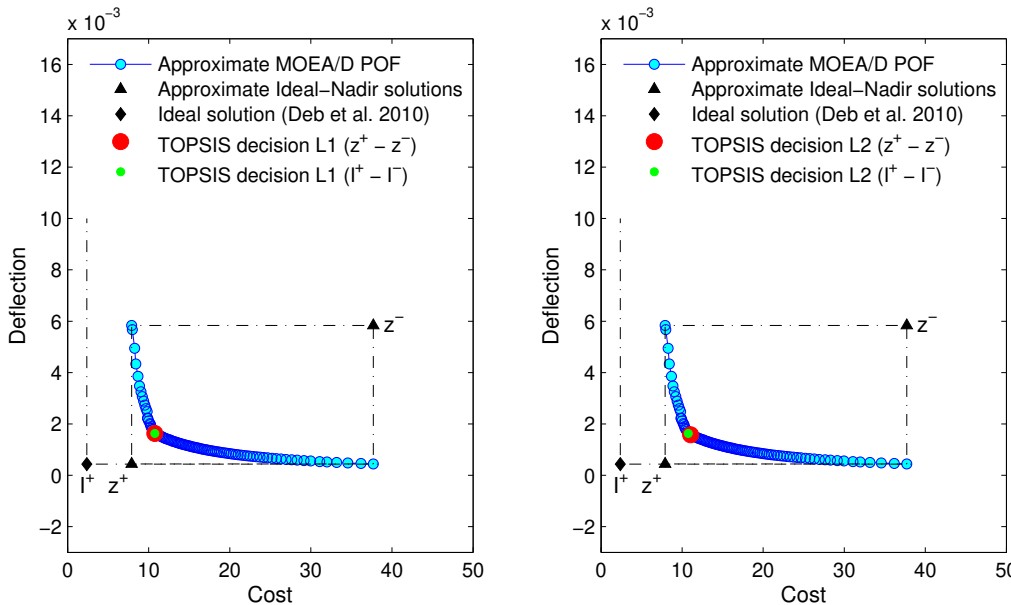

**Figure 16.** TOPSIS decision with $L_1$ (**left**) and $L_2$ (**right**) metrics on the approximate POF with the hypervolume indicator closest to the average value of hypervolume after 100 runs for MOEA/D ($N = 100$).

Finally, when DM's partial-preferences were introduced into the algorithms, the results were very similar to those presented above. In order not to be redundant, only WASF-GA results (DM's partial-preferences: $15, 0.0025$) are shown. The results of the last two columns of Table 6 show that, by using $L_1$ metric in the TOPSIS model, the ranking of the proposal solutions bears the same ranking. However, when $L_2$ metric is used, the ranking of the proposed solutions differs (see the last two columns of Table 7). Figure 17 (left) also shows that the TOPSIS decision (rank-1 solution) does not change when $L_1$ metric is used, and this is not the case when using the metric $L_2$ (Figure 17, right).

**Table 6.** TOPSIS ranking results with $L_1$ metric for WASF-GA, DM's partial-preferences (15, 0.0025) ($N = 100$).

| WASF-GA | Cost | Deflection | $L_1^{z^i z^+}$ | $L_1^{z^i z^-}$ | $S^{z^+ z^-}$ | Rank$^{z^+ z^-}$ | Rank$^{I^+ I^-}$ |
|---|---|---|---|---|---|---|---|
| | 9.3174 | 0.0020 | 0.0076 | 0.0112 | 0.5954 | 1 | 1 |
| | 9.7537 | 0.0019 | 0.0076 | 0.0112 | 0.5954 | 2 | 2 |
| | 9.8696 | 0.0019 | 0.0076 | 0.0112 | 0.5947 | 3 | 3 |
| | 9.1067 | 0.0020 | 0.0076 | 0.0111 | 0.5943 | 4 | 4 |
| | 9.6178 | 0.0019 | 0.0076 | 0.0111 | 0.5935 | 5 | 5 |
| | 10.224 | 0.0018 | 0.0077 | 0.0111 | 0.5919 | 6 | 6 |
| | 8.9470 | 0.0021 | 0.0077 | 0.0111 | 0.5911 | 7 | 7 |
| | 10.314 | 0.0018 | 0.0077 | 0.0112 | 0.5909 | 8 | 8 |

| WASF-GA | Cost | Deflection | $L_1^{z^i I^+}$ | $L_1^{z^i I^-}$ | $S^{I^+ I^-}$ | Rank$^{I^+ I^-}$ | Rank$^{z^+ z^-}$ |
|---|---|---|---|---|---|---|---|
| | 9.3174 | 0.0020 | 0.0213 | 0.9787 | 0.9787 | 1 | 1 |
| | 9.7537 | 0.0019 | 0.0213 | 0.9787 | 0.9787 | 2 | 2 |
| | 9.8696 | 0.0019 | 0.0214 | 0.9786 | 0.9786 | 3 | 3 |
| | 9.1067 | 0.0020 | 0.0214 | 0.9786 | 0.9786 | 4 | 4 |
| | 9.6178 | 0.0019 | 0.0214 | 0.9786 | 0.9786 | 5 | 5 |
| | 10.224 | 0.0018 | 0.0214 | 0.9786 | 0.9786 | 6 | 6 |
| | 8.9470 | 0.0021 | 0.0214 | 0.9786 | 0.9786 | 7 | 7 |
| | 10.314 | 0.0018 | 0.0214 | 0.9786 | 0.9786 | 8 | 8 |

**Table 7.** TOPSIS ranking results with $L_2$ metric for WASF-GA, DM's partial-preferences $(15, 0.0025)$ ($N = 100$).

| WASF-GA | Cost | Deflection | $L_2^{z^i z^+}$ | $L_2^{z^i z^-}$ | $S^{z^+ z^-}$ | Rank$^{z^+ z^-}$ | Rank$^{I^+ I^-}$ |
|---|---|---|---|---|---|---|---|
| | 9.7537 | 0.0019 | 0.0078 | 0.0120 | 0.6054 | 1 | 2 |
| | 9.8696 | 0.0019 | 0.0078 | 0.0119 | 0.6048 | 2 | 5 |
| | 9.6178 | 0.0019 | 0.0080 | 0.0122 | 0.6035 | 3 | 3 |
| | 9.3174 | 0.0020 | 0.0083 | 0.0126 | 0.6034 | 4 | 1 |
| | 9.1067 | 0.0020 | 0.0086 | 0.0129 | 0.6034 | 5 | 4 |
| | 10.224 | 0.0018 | 0.0077 | 0.0115 | 0.6004 | 6 | 7 |
| | 10.118 | 0.0018 | 0.0078 | 0.0116 | 0.5986 | 7 | 9 |
| | 10.314 | 0.0018 | 0.0077 | 0.0115 | 0.5985 | 8 | 10 |

| WASF-GA | Cost | Deflection | $L_2^{z^i I^+}$ | $L_2^{z^i I^-}$ | $S^{I^+ I^-}$ | Rank$^{I^+ I^-}$ | Rank$^{z^+ z^-}$ |
|---|---|---|---|---|---|---|---|
| | 9.3174 | 0.0020 | 0.0213 | 0.9787 | 0.9787 | 1 | 4 |
| | 9.7537 | 0.0019 | 0.0214 | 0.9787 | 0.9786 | 2 | 1 |
| | 9.6178 | 0.0019 | 0.0214 | 0.9786 | 0.9786 | 3 | 3 |
| | 9.1067 | 0.0020 | 0.0214 | 0.9786 | 0.9786 | 4 | 5 |
| | 9.8696 | 0.0019 | 0.0214 | 0.9786 | 0.9786 | 5 | 2 |
| | 8.9470 | 0.0021 | 0.0215 | 0.9786 | 0.9785 | 6 | 9 |
| | 10.224 | 0.0018 | 0.0215 | 0.9786 | 0.9785 | 7 | 6 |
| | 8.8117 | 0.0021 | 0.0215 | 0.9786 | 0.9785 | 8 | 10 |

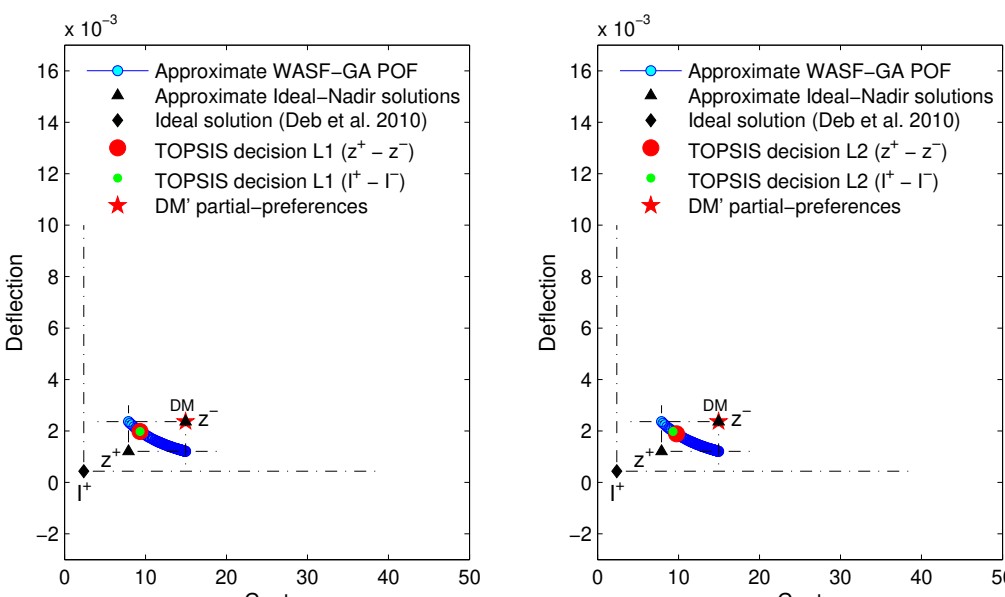

**Figure 17.** TOPSIS decision with $L_1$ (**left**) and $L_2$ (**right**) metrics on the approximate POF with the hypervolume indicator closest to the average value of hypervolume after 100 runs for WASF-GA, DM's partial-preferences $(15, 0.0025)$ ($N = 100$).

### 4.2. Three-Objective Welded Beam Design Problem (Decision)

In this section, problem (18) is redefined considering normal stress $\sigma(\mathbf{x})$ as a third objective function to be minimized. The new mathematical description of the problem [54] is formulated below. By including normal stress as a third objective, the decision-making process in the objective space is even more difficult (see Figure 18).

$$\begin{aligned}
min. \quad & f_1(\mathbf{x}) = 1.10471h^2l + 0.04811tb(14.0 + l) \\
min. \quad & f_2(\mathbf{x}) = \delta(\mathbf{x}) = \frac{2.1952}{t^3b} \\
min. \quad & f_3(\mathbf{x}) = \sigma(\mathbf{x}) = \frac{504000}{t^2b} \\
s.t. \quad & g_1(\mathbf{x}) = 13600 - \tau(\mathbf{x}) \geq 0 \\
& g_2(\mathbf{x}) = 30000 - \sigma(\mathbf{x}) \geq 0 \\
& g_3(\mathbf{x}) = b - h \geq 0 \\
& g_4(\mathbf{x}) = P_c(\mathbf{x}) - 6000 \geq 0 \\
& h, b \in [0.125, 5] \\
& l, t \in [0.1, 10]
\end{aligned}$$

(19)

*where*

$$\tau(\mathbf{x}) = \sqrt{\frac{(\tau'(\mathbf{x}))^2 + (\tau''(\mathbf{x}))^2 + l\tau'(\mathbf{x})\tau''(\mathbf{x})}{\sqrt{0.25(l^2 + (h+t)^2)}}}$$

$$\tau'(\mathbf{x}) = \frac{6000}{\sqrt{2}hl}$$

$$\tau''(\mathbf{x}) = \frac{6000(14 + 0.5l)\sqrt{0.25(l^2 + (h+t)^2)}}{2\left[0.707hl(l^2/12 + 0.25(h+t)^2)\right]}$$

$$P_c(\mathbf{x}) = 64746.022(1 - 0.0282346t)tb^3$$

In this problem, the methodology proposed in this paper was implemented in the following way. After a set of Pareto-optimal solutions was obtained by a MOEA (a set of potential solutions obtained in a randomized trial of NSGA-II with $N = 50$ and $G = 500$ was used for comparisons, see Figure 18), the TOPSIS and ELECTRE I methodologies were used to determine the most attractive solution for a DM. The results shown in Tables 8 and 9 do not differ much from those obtained for the problem (18) with two objective functions. With respect to both the ideal $z^+$ and nadir $z^-$ solutions and the $I^+$ and $I^-$ solutions, the ranking of solutions does not change if the $L_1$ metric is used in the TOPSIS method; this cannot be stated for using $L_2$ metric.

**Table 8.** TOPSIS ranking results with $L_1$ metric for NSGA-II.

| NSGA-II | Cost | Deflection | Stress | $L_1^{z^iz^+}$ | $L_1^{z^iz^-}$ | $S^{z^+z^-}$ | Rank$^{z^+z^-}$ | Rank$^{I^+I^-}$ |
|---------|------|-----------|--------|---------|---------|---------|---------|---------|
| | 23.7093 | 0.0007 | 1596.7010 | 0.0281 | 0.2655 | 0.9043 | 1 | 1 |
| | 28.0821 | 0.0006 | 1338.5632 | 0.0290 | 0.2646 | 0.9013 | 2 | 2 |
| | 19.1353 | 0.0009 | 2007.9248 | 0.0291 | 0.2645 | 0.9010 | 3 | 3 |
| | 22.9850 | 0.0008 | 1734.4095 | 0.0294 | 0.2642 | 0.9000 | 4 | 4 |
| | 26.5692 | 0.0007 | 1476.5583 | 0.0295 | 0.2641 | 0.8997 | 5 | 5 |
| | 26.1854 | 0.0007 | 1507.1893 | 0.0295 | 0.2641 | 0.8996 | 6 | 6 |
| | 17.2381 | 0.0010 | 2235.0525 | 0.0302 | 0.2634 | 0.8971 | 7 | 7 |
| | 30.7015 | 0.0006 | 1260.3816 | 0.0306 | 0.2630 | 0.8957 | 8 | 8 |
| **NSGA-II** | **Cost** | **Deflection** | **Stress** | $L_1^{z^iI^+}$ | $L_1^{z^iI^-}$ | $S^{I^+I^-}$ | Rank$^{I^+I^-}$ | Rank$^{z^+z^-}$ |
| | 23.7093 | 0.0007 | 1596.7010 | 0.0291 | 0.9609 | 0.9706 | 1 | 1 |
| | 28.0821 | 0.0006 | 1338.5632 | 0.0300 | 0.9600 | 0.9697 | 2 | 2 |
| | 19.1353 | 0.0009 | 2007.9248 | 0.0301 | 0.9599 | 0.9696 | 3 | 3 |
| | 22.9850 | 0.0008 | 1734.4095 | 0.0304 | 0.9596 | 0.9693 | 4 | 4 |
| | 26.5692 | 0.0007 | 1476.5583 | 0.0305 | 0.9595 | 0.9692 | 5 | 5 |
| | 26.1854 | 0.0007 | 1507.1893 | 0.0305 | 0.9595 | 0.9692 | 6 | 6 |
| | 17.2381 | 0.0010 | 2235.0525 | 0.0312 | 0.9588 | 0.9684 | 7 | 7 |
| | 30.7015 | 0.0006 | 1260.3816 | 0.0317 | 0.9583 | 0.9680 | 8 | 8 |

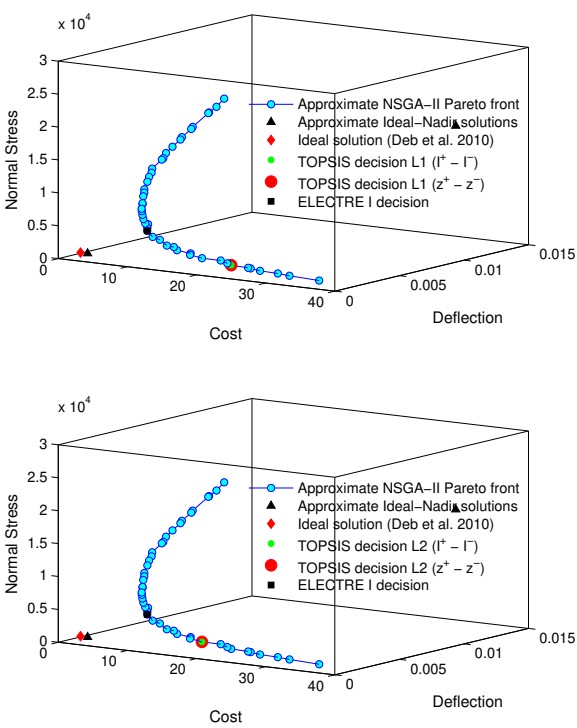

**Figure 18.** TOPSIS decision with $L_1$ (top) and $L_2$ (bottom) metrics and ELECTRE I decision on the approximate POF achieved in a random run for NSGA-II ($N = 50$).

**Table 9.** TOPSIS ranking results with $L_2$ metric for NSGA-II.

| NSGA-II | Cost | Deflection | Stress | $L_2^{z^i z^+}$ | $L_2^{z^i z^-}$ | $S^{z^+ z^-}$ | Rank$^{z^+ z^-}$ | Rank$^{I^+ I^-}$ |
|---|---|---|---|---|---|---|---|---|
| | 19.1353 | 0.0009 | 2007.9248 | 0.0339 | 0.3584 | 0.9137 | 1 | 1 |
| | 17.2381 | 0.0010 | 2235.0525 | 0.0344 | 0.3541 | 0.9115 | 2 | 2 |
| | 23.7093 | 0.0007 | 1596.7010 | 0.0371 | 0.3660 | 0.9080 | 3 | 6 |
| | 22.9850 | 0.0008 | 1734.4095 | 0.0369 | 0.3633 | 0.9077 | 4 | 4 |
| | 21.7416 | 0.0009 | 1948.9974 | 0.0370 | 0.3590 | 0.9065 | 5 | 5 |
| | 17.1718 | 0.0011 | 2418.4668 | 0.0371 | 0.3505 | 0.9044 | 6 | 3 |
| | 15.0645 | 0.0011 | 2599.6250 | 0.0379 | 0.3472 | 0.9017 | 7 | 7 |
| | 26.1854 | 0.0007 | 1507.1893 | 0.0407 | 0.3675 | 0.9003 | 8 | 8 |
| NSGA-II | Cost | Deflection | Stress | $L_2^{z^i I^+}$ | $L_2^{z^i I^-}$ | $S^{I^+ I^-}$ | Rank$^{I^+ I^-}$ | Rank$^{z^+ z^-}$ |
| | 19.1353 | 0.0009 | 2007.9248 | 0.0353 | 0.9649 | 0.9647 | 1 | 1 |
| | 17.2381 | 0.0010 | 2235.0525 | 0.0357 | 0.9637 | 0.9643 | 2 | 1 |
| | 17.1718 | 0.0011 | 2418.4668 | 0.0382 | 0.9613 | 0.9617 | 3 | 6 |
| | 22.9850 | 0.0008 | 1734.4095 | 0.0386 | 0.9647 | 0.9615 | 4 | 4 |
| | 21.7416 | 0.0009 | 1948.9974 | 0.0386 | 0.9628 | 0.9615 | 5 | 5 |
| | 23.7093 | 0.0007 | 1596.7010 | 0.0388 | 0.9661 | 0.9614 | 6 | 3 |
| | 15.0645 | 0.0011 | 2599.6250 | 0.0388 | 0.9610 | 0.9611 | 7 | 7 |
| | 26.1854 | 0.0007 | 1507.1893 | 0.0425 | 0.9648 | 0.9578 | 8 | 8 |

Finally, a study using the ELECTRE I method is included in this section. In a first step, all data (see Columns 1–3 of Table 8 or Table 9 showing eight values out of fifty) were normalized, and equal weights values were assigned to all objective functions. Then, the concordance and discordance coefficients for all the pairs of solutions, according to the authors of [7,16], were calculated to obtain the concordance matrix and discordance matrix. To finish, the aggregate dominance matrix ($50 \times 50$) (Table 10) was determinate by setting the threshold $\bar{c}$ for the concordance test to 0.1 and the threshold $\bar{d}$ for the non-discordance test to 0.9. From the results in Table 10, it can be said that Solution 33 (9.22674, 0.00194, 4446.90136) (cost, deflection and normal stress, respectively) is better than all the

others. A sensitivity analysis of the $\bar{c}$ and $\bar{d}$ values was carried out, and it was found that Solution 33 (9.226740, 0.001943, 4446.901367) is always represented (see Table 11).

**Table 10.** Aggregate dominance matrix ($\bar{c}$ = 0.1, $\bar{d}$ = 0.9).

| Solutions | |
| --- | --- |
| 1 | 0000000000000000000000000000000000000000000000000000 |
| 2 | 0000000000000000000000000000000000000000000000000000 |
| 3 | 1100000000000000000000000000000000000000000000000000 |
| 4 | 1100000000000000000000000000000000000000000000000000 |
| 5 | 1111000000000000000000000000000000000000000000000000 |
| 6 | 1111100000000000000000000000000000000000000000000000 |
| 7 | 1111110000000000000000000000000000000000000000000001 |
| 8 | 1111111000000000000000000000000000000000000000000001 |
| 9 | 1111111100000000000000000000000000000000000000000011 |
| 10 | 1111111110000000000000000000000000000000000000000011 |
| 11 | 1111111111000000000000000000000000000000000000000111 |
| 12 | 1111111111000000000000000000000000000000000000000111 |
| 13 | 1111111111110000000000000000000000000000000000000111 |
| 14 | 1111111111111000000000000000000000000000000000000111 |
| 15 | 1111111111111100000000000000000000000000000000001111 |
| 16 | 1111111111111110000000000000000000000000000000011111 |
| 17 | 1111111111111111000000000000000000000000000000111111 |
| 18 | 1111111111111111100000000000000000000000000001111111 |
| 19 | 1111111111111111110000000000000000000000000011111111 |
| 20 | 1111111111111111111000000000000000000000000111111111 |
| 21 | 1111111111111111111100000000000000000000000111111111 |
| 22 | 1111111111111111111110000000000000000000001111111111 |
| 23 | 1111111111111111111111000000000000000000001111111111 |
| 24 | 1111111111111111111111100000000000000001111111111111 |
| 25 | 1111111111111111111111110000000000000001111111111111 |
| 26 | 1111111111111111111111111000000000000001111111111111 |
| 27 | 1111111111111111111111111110000000000011111111111111 |
| 28 | 1111111111111111111111111111000000000011111111111111 |
| 29 | 1111111111111111111111111111100000001111111111111111 |
| 30 | 1111111111111111111111111111101000011111111111111111 |
| 31 | 1111111111111111111111111111100000001111111111111111 |
| 32 | 1111111111111111111111111111111100011111111111111111 |
| 33 | 1111111111111111111111111111111101111111111111111111 |
| 34 | 1111111111111111111111111111111110011111111111111111 |
| 35 | 1111111111111111111111111100000001111111111111111111 |
| 36 | 1111111111111111111111110000000000111111111111111111 |
| 37 | 1111111111111111111111100000000000011111111111111111 |
| 38 | 1111111111111111111110000000000000011111111111111111 |
| 39 | 1111111111111111110000000000000000001111111111111111 |
| 40 | 1111111111111111110000000000000000001011111111111111 |
| 41 | 1111111111111111000000000000000000000111111111111111 |
| 42 | 1111111111111000000000000000000000000011111111111111 |
| 43 | 1111111111110000000000000000000000000001111111111111 |
| 44 | 1111111111110000000000000000000000000000111111111111 |
| 45 | 1111111100000000000000000000000000000000011111111111 |
| 46 | 1111111100000000000000000000000000000000001111111111 |
| 47 | 1111111000000000000000000000000000000000000111111111 |
| 48 | 1111110000000000000000000000000000000000000011111111 |
| 49 | 1100000000000000000000000000000000000000000000000001 |
| 50 | 0000000000000000000000000000000000000000000000000000 |

**Table 11.** Sensitivity analysis to variations in the thresholds $\bar{c}$ and $\bar{d}$.

| $\bar{c}$ | $\bar{d}$ | Solutions |
|-----------|-----------|-----------|
| 0.1 | 0.9 | 33 |
| 0.2 | 0.8 | 33–34 |
| 0.33 | 0.67 | 28–29–33–34 |

To conclude, the values of the solutions calculated by TOPSIS and ELECTRE I are shown numerically in Table 12 and graphically in Figure 18. Table 12 shows that ELECTRE I obtained a lower value of the cost function than obtained by TOPSIS. However, lower deflection and normal stress values were achieved by TOPSIS. Besides, with the $L_1$ metric, as demonstrated in Section 3, TOPSIS guarantees that the proposed solution is classified with respect to both the ideal $I^+$ and nadir $I^-$ solutions even if these are not known. It could also be deduced from the results in Figure 18 that the solution achieved by ELECTRE I resides or is close to a knee region [62–68] where a small improvement in one of the objectives leads to a significant degradation in at least one of the other objectives, and therefore it may be of more interest to a DM than the solution calculated by TOPSIS. In any case, the selection of the best MCDM method for a given problem can be a difficult task [69], and it is not within the scope of this work.

**Table 12.** Results for the $\text{TOPSIS}^{L_1}$ and ELECTRE I methods.

|  | Cost | Deflection | Normal Stress |
|---|------|-----------|---------------|
| $\text{TOPSIS}^{L_1}$ decision | 23.709299 | 0.000696 | 1596.701050 |
| ELECTRE I decision | 9.226740 | 0.001943 | 4446.901367 |

## 5. Conclusions

Usually in the literature of Multi-objective Metaheuristics (MOMH), the background on Multiple Criteria Decision-Making is ignored. This work is so rich not only in the classical aspects of MOMH but also in the MCDM. In this context, this paper proposes and demonstrates the effectiveness of a search procedure that brings together two independent technical stages of MOO and MCDM.

In the optimization stage, a variety of representative a posteriori algorithms, NSGA-II, GWASF-GA and MOEA/D, and with DM's partial-preferences, g-NSGA-II and WASF-GA, were used during the optimization process, in order to obtain an approximate Pareto-optimal Front. An original comparison of results based on hypervolume metric were performed on a welded beam engineering design referent problem (two objective functions). This problem is characterized because there is no knowledge of the ideal and nadir solutions. The obtained results clearly indicate that the NSGA-II and GWASF-GA algorithms achieved similar and better performances than those obtained by the MOEA/D algorithm. In addition, NSGA-II and GWASF-GA obtained the best results compared to other metheuristic methods of the literature. When partial preferences were introduced into the algorithms, the results of the comparisons between g-NSGA-II and WASF-GA were different depending on the used reference point (DM's partial-preferences). When the reference point was set to $(15, 0.0025)$ (feasible) (towards the POF area corresponding to well-balanced solutions), the best result was obtained by WASF-GA. When the reference point was $(30, 0.001)$ (feasible), g-NSGA-II and WASF-GA achieved similar results, although g-NSGA-II had a slightly better performance of the hypervolume metric and better distribution of the approximate Pareto solution set. Finally, when the reference point was set to $(4, 0.003)$ (no-feasible), the performances were similar for both algorithms.

In the decision analysis stage, the TOPSIS methodology is proposed. Although this method requires knowledge of the ideal and nadir solutions of the MOP, in this work only approximate Pareto-optimal fronts are studied and, therefore, the ideal and nadir solutions may not be known. However, in this paper, it is shown that, by using the $L_1$ distance metric in the TOPSIS method, the classification of the proposed solutions has the same range. This is valid with respect to both the

ideal and nadir solutions of the approximate POF obtained by the algorithms, and the true ideal and nadir solutions of the POM. This cannot be stated by using $L_2$ metric. For demonstration, a comparison of $L_1$ and $L_2$ metrics in TOPSIS model was performed for the studied bi-objective welded beam problem. Finally, a comparison (three objective functions) of the solutions proposed by TOPSIS and ELECTRE I was carried out. The results show that a lower value of the cost function was obtained by ELECTRE I. However, lower deflection and normal stress values were achieved by TOPSIS.

In our opinion, the proposed methodology in this work is a suggestive method for problems similar to the one studied in this paper and it may be a useful tool and provide an important clue to a DM in his/her final decision.

**Author Contributions:** All authors have contributed equally to the realization of this work. M.M., M.F., F.M. and R.A.-C. participated in the conception and design of the work; M.M., M.F., F.M. and R.A.-C. reviewed the bibliography; M.M., M.F., F.M. and R.A.-C. conceived and designed the experiments; M.M., M.F., F.M. and R.A.-C. performed the experiments; M.M., M.F., F.M. and R.A.-C. analyzed the data; and M.M., M.F., F.M. and R.A.-C. wrote and edited the paper. All authors have read and agreed to the published version of the manuscript.

**Funding:** This research received no external funding.

**Acknowledgments:** This work was possible thanks to the collaboration and support of the University Institute of Intelligent Systems and Numeric Applications in Engineering (IUSIANI-ULPGC).

**Conflicts of Interest:** The authors declare no conflict of interest.

## Abbreviations

The following abbreviations are used in this manuscript:

| | |
|---|---|
| MOP | Multi-Objective Optimization Problem |
| MCDM | Multiple Criteria Decision-Making |
| MOO | Multi-Objective Optimization |
| MOMH | Multi-Objective Metaheuristic |
| DM | Human Decision-Maker |
| POF | Pareto-Optimal front |
| MOEA | Multi-Objective Evolutionary Algorithm |
| GFCL | Generate First–Choose Later |
| NSGA-II | Non-Dominated Sorting Genetic Algorithm-II |
| MOEA/D | Multi-Objective Evolutionary Algorithm based on Decomposition |
| GWASF-GA | Global Weighting Achievement Scalarizing Function Genetic Algorithm |
| WASF-GA | Weighting Achievement Scalarizing Function Genetic Algorithm |
| g-NSGA-II | Non-g-Dominated Sorting Genetic Algorithm |
| NFEs | Number of Function Evaluations |
| TOPSIS | Technique for Order Preference by Similarity to an Ideal Solution |
| ELECTRE | ELimination Et Choix Traduisant la REalité |
| ODEMO | Orthogonal Differential Evolution for Multiobjective Optimization |
| MOWCA | Multi-Objective Water Cycle Algorithm |
| M2O-CSA | Multi-Objective Orthogonal Opposition-Based Crow Search Algorithm |
| MOCSA | Multi-Objective Crow Search Algorithm |
| MOCCSA | Multi-Objective Chaotic Crow Search Algorithm |
| ANN | Artificial Neural Network |
| MOMPC | Multi-objective Model Predictive Control |

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
