# Peer review of "TOPSIS Decision on Approximate Pareto Fronts by Using Evolutionary Algorithms: Application to an Engineering Design Problem"

_mathematics, doi:10.3390/math8112072_

Round 1
Reviewer 1 Report
This work proposed a two stage multi-objective optimisation method where at the second stage the best model will be selected from a front of solutions evolved from the first stage. This is accomplished by the new TOPSIS method. The idea is interesting but somehow similar with the well known ‘knee point’ in multi-objective research. I would like the authors to mention and explain on this point. The paper is generally clear and well organised. However, I do have some more concerns. Detailed comments:- The first paragraph in Page 2 only have one sentence. I think it is a formatting issue.
- You have introduced a lot on MOO but actually the main contribution of this work is on MCDM, so please add more details on the review on reference[45-52].
- “potential MOEA” is confusing.
- For real-world problem how to find I+ and I- in Equation (8)
- The description of the Welded beam problem does not show that it has two conflicting objectives. Please make this point clear.
- There are many grammar issues in the paper, please do have a detailed check and revise it.
Reviewer 2 Report
The Authors proposed a novel method of ordering nondominated solutions of MOP. The aim of the method is to help the decision maker.
In the introduction, three commonly used MOP algorithms are briefly described: NSGA-II, MOEA/D and WASF-GA. In my opinion Authors did not explained why they chose these methods, but not. i.e. SPEA-2. On the other hand, optimization algorithm in the first stage of the proposed methodology can be changed easily.
However MOP methods are described sufficiently, MCDM methods other than TOPSIS are not mentioned. There are many methods for supporting DM such as Best Worst Method, Goal Programming, Weighted Product/Sum Model etc. Why TOPSIS was chosen?
Basic concepts are stated correctly. Figure 1 is illustrative.
Methodology is presented clearly. Proposition and proof are correct.
Results section: The authors should provide the formula for calculating hypervolume. It is obvious for readers familiar with MOEA but some readers can be uncertain what it is.
Results are presented clearly. Discussion is comprehensive.
References are sufficient. About 15 out of 61 works cited were published in last 5 years.
Reviewer 3 Report
1. Paper summary
To solve the multi-objective optimization problems, This paper proposes a methodology that follows a two-stage MOO+MCDM procedure. In the MOO stage (GF), a potential MOEA obtains an approximate POF of solutions. The obtained results clearly indicated that the NSGA-II and the GWASF-GA algorithms achieved similar and better performances than those obtained by the MOEA/D algorithm. Then, in the MCDM stage (CL), the L1 distance metric is proposed and used in TOPSIS method in order automatically to obtain an approximate ranking of the solutions that could be interesting to a DM. A lower deflection and normal stress values were achieved by TOPSIS.
2. Major comments
There were two and three criteria for comparison at the stage of optimization stage and decision analysis stage, respectively. In this paper, proposed method shows good performance compared to the algorithms, but if the number of criteria is increased or changed, it is unclear whether the superiority of the proposed method is maintained or not. In other words, there is a risk of becoming specific to a particular situation. Therefore, it is difficult to ensure that non-dominated solutions can be obtained as the criteria for multi-objective optimizations are increased.
3. Minor comments
3-1. Express the flow of your proposal method in an algorithmic way. Although I can check the overview in Figure 2 of two-stage methodology, it is recommended to make it a little more intuitive to know how to use the (1)-(17) equation definition in your proposed method.
3-2. Need a kinder explanation of motivation. For example, descriptions based on related works can be supplemented. You mentioned that it is not necessary to know the ideal solution and the nadir solution in the proposed method. In this context, it is suggested to provide a supplementary explanation following the related works.
4. Typo and semantic comments
1) Spaces is required.
(1) Sentence: ...some criteria.The Non-Dominated...
-> ...criteria. The...
2) Spaces is needed to reduce
(1) Sentence: ....11 design benchmark problem. The problem was solved with....
-> ...problem. The...
(2) Sentence: The best known ideal and nadir values [2.3810, 0.000439] and [333.9095, 0.0713] respectively [53] of the problem (18) have been used in the experiments, see Figure 11 (right) .
-> ...(right).
2) Sentence: Keywords: multiple criteria decision making; TOPSIS; preferences; engineering design; optimization; 14 multi-objective evolutionary algorithm
-> Please consider following sentence: ‘decision making’ to ‘decision-making’
3) Sentence: These algorithms does not guarantee the determination of the exact POF as any heuristic does but the result is very close to the exact solution.
-> Please consider following sentence: ‘does’ to ‘do’
4) Sentence: The first, known as multiple criteria decision making (MCDM) [1–12], is essentially interested in decision making
-> Please consider following sentence: ‘decision making’ to ‘decision-making’
5) Sentence: the Multi-Objective Evolutionary Algorithm based on Decomposition (MOEA/D) [33] and the Global Eeighting Achievement Scalarizing Function Genetic Algorithm (GWASF-GA) [34] are recognized algorithms in the multi-objective literature that use this approach.
-> Please consider following sentence: ‘Eeighting’ to ‘Weighting’
6) Sentence: MOEAs have extensive applications in the engineering field [44] and some of them, propose a two-stage methodologie.
-> Please consider following sentence: ‘methodologie’ to ‘methodologies’ or ‘methodology’
7) Sentence: This problem is characterised by a lack of knowledge about ideal and nadir values [53].
-> Please consider following sentence: ‘characterised’ to ‘characterized’
8) Sentence: Formally, the bi-objective welded beam design problem can be de defined as follows:
-> Please consider following sentence: Whether 'de' be eliminated or not
9) Sentence: Also, in Figure 5 (left) that presents the evolution of the average hypervolume per generation and in Figure 5 (write) which shows the evolution of the standard deviation hypervolume
-> Please consider following sentence: ‘(write)’ to ‘(right)’
Round 2
Reviewer 1 Report
The authors have addressed most of my concerns. I am happy with the revision.
Reviewer 3 Report
The paper is revised thoroughly according to the comments of the reviewer. This can be recommended to be accepted.